# Anticipatory Music Transformer

**John Thickstun**                                                                  *jthickstun@cs.stanford.edu*
*Department of Computer Science*
*Stanford University*

**David Hall**                                                                      *dlwh@stanford.edu*
*Department of Computer Science*
*Stanford University*

**Chris Donahue**                                                                   *chrisdonahue@cmu.edu*
*Google DeepMind and Carnegie Mellon University*

**Percy Liang**                                                                     *pliang@cs.stanford.edu*
*Department of Computer Science*
*Stanford University*

**Reviewed on OpenReview:** *https://openreview.net/forum?id=EBNJ33Fcrl*

## Abstract

We introduce *anticipation*: a method for constructing a controllable generative model of a temporal point process (the event process) conditioned asynchronously on realizations of a second, correlated process (the control process). We achieve this by interleaving sequences of events and controls, such that controls appear following stopping times in the event sequence. This work is motivated by problems arising in the control of symbolic music generation. We focus on infilling control tasks, whereby the controls are a subset of the events themselves, and conditional generation completes a sequence of events given the fixed control events. We train anticipatory infilling models using the large and diverse Lakh MIDI music dataset. These models match the performance of autoregressive models for prompted generation, with the additional capability to perform infilling control tasks, including accompaniment. Human evaluators report that an anticipatory model produces accompaniments with similar musicality to even music composed by humans over a 20-second clip.

## 1 Introduction

Imagine you are given a melody, and asked to compose a harmonizing accompaniment. This melody can be modeled by a *temporal point process*: a probability distribution over (musical) events that arrive stochastically at points in time. An accompaniment to this melody is also a realization of a temporal point process. The events in the accompaniment are tightly correlated—but often asynchronous—with the events in the melody. Generating an accompaniment to a given melody is an example of a control task: we seek the ability to generate an accompaniment (the events) conditioned on a given melody (the controls). Models that generate symbolic music (i.e., compose) subject to user-specified controls are of broad interest as tools for music co-creation (Louie et al., 2020).

Motivated by this example, we are interested in constructing generative models of a temporal point process (the *event* process) that can be conditioned on realizations of a second, correlated point process (the *control* process). Generating an accompaniment to a melody is an instance of a more general *infilling* task, whereby we generate a complete realization of a temporal point process given partial observation of a subset of its events. Infilling is a powerful control mechanism for music generation: previous work on musical infilling (Huang et al., 2017) powered the J.S. Bach Google Doodle (Huang et al., 2019), an interactive music experience with broad popular appeal.

The dynamics of a temporal point process can be captured by a neural autoregressive model trained to predict the next event in a time-ordered sequence (Du et al., 2016). A natural extension of this paradigm to conditional distributions is sequence-to-sequence modeling (Seq2Seq) (Sutskever et al., 2014), whereby the control sequence is prepended to the sequence of events. For long sequences, Seq2Seq places time-localized controls far from the events they describe. While there is substantial recent work on long-context modeling (Child et al., 2019; Dao et al., 2022; Gu et al., 2022; Hawthorne et al., 2022), rather than brute-force the learning of artificial long-range dependencies, we propose to structure conditional generation so that a control on time $t$ is located close to events near time $t$. Our premise is that the most relevant context for predicting the next event is the recent event history (unidirectional context) and both recent and near-future controls (bidirectional context).

Standard practice to efficiently train an autoregressive model relies on the observation that context for prediction at one index in the sequence is a prefix of the context for predictions at future indices. This allows us to update the model based on $M - 1$ predictions for each sequence of length $M$. Conditioning on asynchronous controls by constructing an ad-hoc context (e.g., the $M/2$ previous events and $M/2$ nearest controls) to predict each event would be computationally wasteful: we want to define a single, coherent interleaved sequence of events and controls. This is straightforward if the events and controls are synchronous: to condition on bidirectional control context $[t - \delta, t + \delta]$ at time $t$, simply shift the control sequence by a time offset $\delta$ and construct a joint sequence by interleaving events and controls at alternating sequence positions. Or alternatively, construct an encoder-decoder model that ingests the paired control tokens through a separate encoder (Raffel et al., 2020).

When events and controls are asynchronous, simple approaches to interleaving these sequences make sampling from the ensuing joint model intractable. This includes the natural *sort order*, whereby we interleave a control on time $t$ as if it were at time $t - \delta$. For tractable sampling, we will see in Section 3 that the index in the interleaved sequence that immediately precedes a control must be a *stopping time* (Billingsley, 1995). We propose a method for interleaving asynchronous events and controls such that a control on time $t$ appears in the interleaved sequence at a stopping time close to events near time $t - \delta$. We call this method *anticipation*. The interval $\delta > 0$ is a hyperparameter chosen to be long enough to give the model time to account for (i.e. anticipate) upcoming controls, but short enough to maintain proximity of events and controls (if $\delta = \infty$, we recover Seq2Seq modeling). The interleaved structure of anticipation is visualized in Figure 1.

**Contributions.** We define an arrival-time encoding of events and controls that is amenable to expressive autoregressive sequence modeling and facilitates anticipation (Section 2). We describe the interleaved structure of an anticipatory autoregressive model, and how to train and sample from this model (Section 3). We apply anticipation to construct anticipatory infilling models for music, trained on the Lakh MIDI music dataset (Raffel, 2016). These models unlock new control capabilities for music generation without sacrificing the performance of unconditional generation (Section 4). We release all code for reproducing these models, along with pre-trained model weights.[1]

## 2 Music as a Temporal Point Process

A marked temporal point process is a probability distribution over sparse events situated at points in continuous time (Daley & Vere-Jones, 2007).

**Definition 2.1.** A *marked temporal point process* is a probability distribution over events $\mathbf{e}_i = (t_i, \mathbf{m}_i)$, where $t_i \in \mathbb{R}_+$ ($t_i \leq t_j$ if $i < j$) is a point in time and $\mathbf{m}_i \in \mathcal{V}$ is a mark from a finite vocabulary $\mathcal{V}$.

Given controls $\mathbf{u}_{1:K}$ provided by a user, we say that we can *control* generation of the events $\mathbf{e}_{1:N}$ with respect to $\mathbf{u}_{1:K}$ if we can sample from $p(\mathbf{e}_{1:N}|\mathbf{u}_{1:K})$. We focus on *infilling* control, whereby the controls $\mathbf{u}_{1:K}$ share a vocabulary with the events $\mathbf{e}_{1:N}$. Given a user-specified set of $K$ events $\mathbf{u}_{1:K}$, we would like to generate a complete realization of the process $\mathbf{e}_{1:N}$ such that $\mathbf{u}_{1:K} \subseteq \mathbf{e}_{1:N}$. This generalizes the *span-infilling* task—which asks us to generate a missing contiguous span of events—previously studied in the music literature (Ippolito et al., 2018; Chang et al., 2021; Tan et al., 2022a;b).

---

[1]For assets and supplemental material, see: https://johnthickstun.com/anticipation/

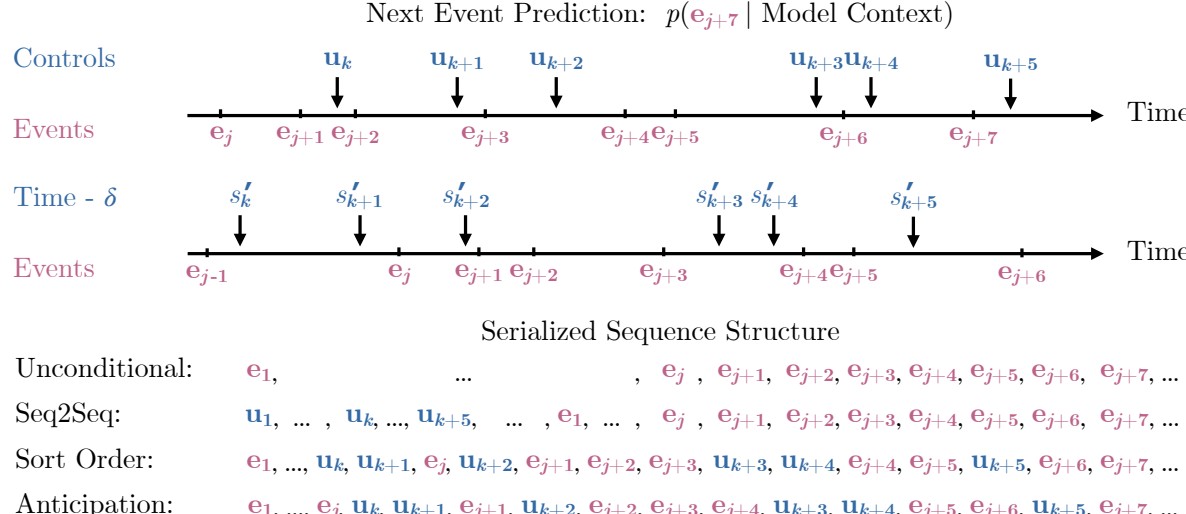

Figure 1: We construct generative models for sequences of events $\mathbf{e}_{1:N}$, conditioned on controls $\mathbf{u}_{1:K}$. We serialize these paired sequences to define an autoregressive factorization of the joint distribution over events and controls. Anticipation interleaves event and control sequences so that a control $\mathbf{u}_k$ on time $s_k$ appears in the recent history when predicting events near time $s_k$. An anticipated control $\mathbf{u}_k$ on time $s_k$ appears as if it were at approximately time $s'_k = s_k - \delta$. For example, when predicting $\mathbf{e}_{j+7}$ above, the recent context of the anticipation sequence contains contains past events and controls, as well as the future control $\mathbf{u}_{k+5}$; we say that a model predicting $\mathbf{e}_{j+7}$ given this context *anticipates* the control $\mathbf{u}_{k+5}$, approximately $\delta$ seconds in advance. Crucially, to be able to condition on controls, the index that immediately preceeds each control in the serialized sequence must be a stopping time, a property that naively interleaving events and controls using the sort order of times $s'_k$ does not satisfy.

Music can be described by a marked temporal point process, where the vocabulary of marks consists of musical notes and other musical events. In this work, we restrict the vocabulary of musical events to notes consisting of a *pitch* $\mathbf{p}$, *instrument* class $\mathbf{k}$, and *duration* $\mathbf{d}$. Following the MIDI standard (International MIDI Association, 1988), we quantize pitch $\mathbf{p} \in \{0, \ldots, 127\}$ according to the 12-tone Western scale ($p = 60$ denotes "middle C", or 261Hz); we represent instrument classes $\mathbf{k} \in \{0, \ldots, 128\}$ using standard MIDI instrument codes, including drums (code 128); we represent duration in units of seconds, quantized to 10ms intervals with a maximum duration of 10 seconds. We represent combined pitch and instrument information using a single value $\mathbf{n} = 128\mathbf{k} + \mathbf{p}$. A mark is thus a note $\mathbf{m}_i = (\mathbf{d}_i, \mathbf{n}_i) \in \mathcal{V}$ from a vocabulary of 17,512 marks, and music consists of these notes situated at points in time.

Older work on music generation typically rasterizes time, encoding music as a uniformly sampled matrix or tensor, i.e., a piano-roll (Boulanger-Lewandowski et al., 2012; Dong et al., 2018). In the piano-roll setting, there are strong solutions to accompaniment and infilling tasks that avoid the complexities of asynchronicity that we address in this paper (Hadjeres et al., 2017; Huang et al., 2017; Pati et al., 2019; Chen et al., 2020). However, for diverse and rhythmically intricate music data (including the Lakh MIDI dataset) piano-roll rasterization comes at a cost: if the rasterization is coarse then rhythmic detail is lost, and if the rasterization is fine then the piano-rolls are high-dimensional, presenting computational challenges. We instead chose to model music as a temporal point process, an approach inspired by Huang et al. (2018); Oore et al. (2020).

**Modeling Temporal Point Processes.** Much of the machine learning literature on temporal point processes focuses on modeling conditional intensity functions (Du et al., 2016; Mei & Eisner, 2017; Omi et al., 2019; Zuo et al., 2020). In contrast, we will model the probability distribution over the next event in a time-ordered sequence. In this regard, our work is most comparable to Shchur et al. (2020); but whereas that work models continuous densities, we model discrete distributions over quantized time values. This allows us to directly apply the full modern machinery of causal autoregressive transformers and large language models to modeling point processes.

**Modeling Arrival Times.** Stochastic arrival times $t_i$ are the defining characteristic of a temporal point process. In Section 2.1 we propose an *arrival-time tokenization* that represents a realization of a marked point process as a sequence of (discretized) arrival times and marks. In Section 3 we exploit an invariance of this sequence representation (context-free subsequences) to create anticipation. We also consider an *interarrival-time tokenization* that is comparable to encodings used in recent work on music generation (Huang et al., 2018; Donahue et al., 2019; Payne, 2019; Oore et al., 2020). Both of these encodings reduce point process modeling to sequence density estimation. However, only the proposed arrival-time encoding facilitates the construction of anticipation.

## 2.1 Encoding Music as Sequences

We represent arrival times $t_i$ using a finite vocabulary of 10,000 values, quantizing time in 10ms intervals (the quantization proposed by Music Transformer Huang et al. (2018)) with a maximum time of 100 seconds. While many musical performances are longer than 100 seconds, we relativize arrival times to the beginning of the model context. The models described in this paper have context length $M = 1024$. Differences between $M$ successive arrival times that exceed 100 seconds appear in less than 0.1% of Lakh MIDI training examples; we discard these examples during preprocessing. By quantizing time we reduce realizations of temporal point processes to discrete sequences composed of successive pairs of times $\mathbf{t}_i \equiv \texttt{quantize}(t_i)$ and marks $\mathbf{m}_i$ or, in the music application, triplets of time, duration, and note $(\mathbf{t}_i, \mathbf{d}_i, \mathbf{n}_i)$.

**Definition 2.2.** The *arrival-time tokenization* of events $\mathbf{e}_{1:N}$ is a sequence $\mathbf{x}_{1:3N}$ defined by

$$\mathbf{x}_{3i-2} = \mathbf{t}_i, \qquad\qquad \mathbf{x}_{3i-1} = \mathbf{d}_i, \qquad\qquad \mathbf{x}_{3i} = \mathbf{n}_i. \qquad (1)$$

The vocabulary has size $|\mathcal{V}| = 27{,}512$: 10,000 quantized times, 1000 quantized durations, and 16,512 instrument-pitch pairs.

Crucially, the triplets $(\mathbf{x}_{3i-2}, \mathbf{x}_{3i-1}, \mathbf{x}_{3i})$ of these sequences are *context-free*: if we re-order the triplets, the semantics of the sequence do not change. We can recover the original ordering by sorting these triplets according to their arrival times $\mathbf{x}_{3i-2}$. We exploit the re-orderability of arrival time tokenized sequences to construct anticipatory autoregressive infilling models in Section 3. A similar encoding has been used by Gardner et al. (2021), but their work does not exploit this re-orderability.

The more common practice in symbolic music modeling encodes music as a sequence of *onset* and *offset* events, separated by interarrival times $\Delta \in \mathbb{R}_+$. For each event $\mathbf{e} = (\mathbf{t}, \mathbf{m})$, we define onset $\mathbf{on} = (\mathbf{t}, \mathbf{n})$ and offset $\mathbf{off} = (\mathbf{t} + \mathbf{d}, \mathbf{n})$. We bound the interarrival time between events by 10 seconds, corresponding to a vocabulary of 1000 possible (10ms quantized) interarrival times.

**Definition 2.3.** Given events $\mathbf{e}_{1:N}$, let $\mathbf{x}'_{1:2N}$ denote an interleaved sequence of onsets $\mathbf{on}_{1:N}$ and offsets $\mathbf{off}_{1:N}$, ordered by time, where values $\mathbf{x}'_i = (\mathbf{t}'_i, \mathbf{n}'_i)$ have interarrival times $\Delta_i = \mathbf{t}'_{i+1} - \mathbf{t}'_i$. The *interarrival-time tokenization* of events $\mathbf{e}_{1:N}$ is a sequence $\mathbf{x}_{1:4N}$ defined by

$$\mathbf{x}_{2i-1} = \mathbf{x}'_i, \qquad\qquad\qquad \mathbf{x}_{2i} = \Delta_i. \qquad (2)$$

Following standard practice, we omit interarrival tokens when $\Delta_i = 0$. The vocabulary has size $|\mathcal{V}| = 34{,}024$: 16,512 note onsets, 16,512 note offsets, and 1000 quantized interarrival times.

In contrast to Definition 2.2, the interarrival-time tokenization described by Definition 2.3—and other common music encodings including including REMI (Huang & Yang, 2020) and OctupleMidi (Zeng et al., 2021)—are *context sensitive*: the timings of tokens are determined contextually by their positions in the sequence. Interarrival-time tokenization may appear less compact than arrival-time tokenization: each event is encoded as an onset and an offset—each with an associated interarrival time—totaling 4 tokens, versus a triplet of arrival time, duration, and note totaling 3 tokens. However, omitting zero-duration interarrival times results in sequences of comparable length under either encoding of the Lakh MIDI dataset.

## 3 Anticipation

Given an *anticipation interval* $\delta > 0$, we construct a sequence $\mathbf{a}_{1:N+K} = \texttt{interleave}_\delta(\mathbf{e}_{1:N}, \mathbf{u}_{1:K})$ that interleaves events $\mathbf{e}_{1:N}$ and controls $\mathbf{u}_{1:K}$ such that a control $\mathbf{u}_k$ on time $s_k$ appears close to events near time $s_k - \delta$; i.e., we *anticipate* $\mathbf{u}_k$, $\delta$ seconds in advance. We can then construct a standard autoregressive sequence model over interleaved sequences $\mathbf{a}_{1:N+K}$. Model predictions under the anticipatory ordering $\mathbf{a}_{1:N+K}$ combine a filtering (i.e., causal) estimate based on the local history of events with a smoothing (i.e., bidirectional) estimate based on local controls (Wiener et al., 1949). The value of $\delta$ is a hyperparameter that controls the degree of smoothing. If $\delta$ is too short, then the model will be blind to upcoming control context. If $\delta$ is too long, then time-locality of the sequence $\mathbf{a}_{1:N+K}$ is broken.

The natural *sort order* that interleaves events and controls by merging a control on time $s_k$ between events at times $t_j, t_{j+1}$ such that $t_j \le s_k - \delta < t_{j+1}$ makes inference intractable. During inference, we only have access to prefixes of $\mathbf{a}_{1:N+K}$, and so the criterion that determines whether to condition on $\mathbf{u}_k$ at index $i$ must be a function of the history $\mathbf{a}_{1:i-1}$. However, the sort order placement of a control $\mathbf{u}_k$ depends on both the event that precedes it and the event that *follows* it in the merged sequence, which requires information that is unavailable at inference time. We formalize this problem using the concept of a *stopping time* (Definition 3.3). Definition 3.1 describes a version of anticipation that interleaves controls $\mathbf{u}_k$ at indices $\tau_{\mathbf{u}_k} \in \{1, \ldots, N+K\}$ after a stopping time $\tau_{\mathbf{u}_k} - 1$.

**Definition 3.1.** (Anticipation) Let $\mathbf{e}_{1:N}$ be events with vocabulary $\mathcal{V}$ and let $\mathbf{u}_{1:K}$ be controls with vocabulary $\mathcal{U}$. Given $\delta > 0$, we define a combined sequence $\mathbf{a}_{1:N+K} \equiv \texttt{interleave}_\delta(\mathbf{e}_{1:N}, \mathbf{u}_{1:K})$ with vocabulary $\mathcal{U} \cup \mathcal{V}$ that interleaves these two sequences. We write $t_j, s_k$ to indicate the arrival times of event $\mathbf{e}_j$ and control $\mathbf{u}_k$ respectively. To simplify notation we define $t_0 = s_0 = -\infty$. In the combined sequence $\mathbf{a}_{1:N+K}$, event $\mathbf{e}_j$ and control $\mathbf{u}_k$ appear (respectively) at indices

$$\tau_{\mathbf{e}_j} = j + \underset{0 \le k \le K}{\arg\max}\{t_{j-1} \ge s_k - \delta\}, \tag{3}$$

$$\tau_{\mathbf{u}_k} = k + \underset{0 \le j \le N}{\arg\min}\{t_j \ge s_k - \delta\}. \tag{4}$$

Unpacking this definition, Equation (3) says that event $\mathbf{e}_j$ appears in sequence $\mathbf{a}_{1:N+K}$ after the first $j-1$ events (term $j$ in the sum) and after any controls that appear earlier in sequence $\mathbf{a}_{1:N+K}$ (the arg max term, mirroring Equation (4)). Equation (4) says that control $\mathbf{u}_k$ appears after the first $k-1$ controls and *after* the first event $\mathbf{e}_j$ exceeding time $s_k - \delta$. The indices $\tau$ of events and controls in the sequence $\mathbf{a}_{1:N+K}$ are random variables, determined by stochastic realizations of $\mathbf{e}_{1:N}$ and $\mathbf{u}_{1:K}$.

An *anticipatory autoregressive model* is an autoregressive model defined over sequences of events and controls $\mathbf{a}_{1:N+K}$ interleaved according to Equations (3) and (4):

$$p(\mathbf{a}_{1:N+K}) = \prod_{i=1}^{N+K} p(\mathbf{a}_i | \mathbf{a}_{1:i-1}). \tag{5}$$

**Example 3.2.** Suppose $t_1 = 1$, $t_2 = 3$, $t_3 = 5$, $s_1 = 7$, and $\delta = 5$. Then $s_1 - \delta = 2$ and

$$\tau_{\mathbf{e}_1} = 1 + \underset{k}{\arg\max}\{-\infty \ge s_k - \delta\} = 1, \tag{6}$$

$$\tau_{\mathbf{e}_2} = 2 + \underset{k}{\arg\max}\{1 \ge s_k - \delta\} = 2, \tag{7}$$

$$\tau_{\mathbf{e}_3} = 3 + \underset{k}{\arg\max}\{3 \ge s_k - \delta\} = 4, \tag{8}$$

$$\tau_{\mathbf{u}_1} = 1 + \underset{j}{\arg\min}\{t_j \ge 2\} = 3. \tag{9}$$

Therefore $\mathbf{a}_{1:4} = (\mathbf{e}_1, \mathbf{e}_2, \mathbf{u}_1, \mathbf{e}_3)$. Contrast this order with the more natural sort order that interleaves $\mathbf{u}_1$ as if it were an event at time $s_1 - \delta = 2$: $(\mathbf{e}_1, \mathbf{u}_1, \mathbf{e}_2, \mathbf{e}_3)$. Under the sort order, $\mathbf{u}_1$ appears *before* the first event $\mathbf{e}_j$ exceeding time $s_1 - \delta$, i.e., the event $\mathbf{e}_2$ appearing at index 3. This rule for placing $\mathbf{u}_1$ cannot be

applied during autoregressive inference, as it requires us to place $\mathbf{u}_1$ at index 2 with foreknowledge of the event $\mathbf{e}_2$ that has yet to be generated. Under the anticipatory order (Definition 3.1), $\mathbf{u}_1$ appears *after* the event $\mathbf{e}_2$ appearing at index 2. This rule for placing $\mathbf{u}_1$ can be applied during the process of autoregressive inference: upon observing $\mathbf{e}_2$ at index 2, we place $\mathbf{u}_1$ at index 3. This distinction between the anticipatory order (which admits autoregressive inference) and the sort order (which does not) can be formalized using the concept of a *stopping time*, which we describe in Section 3.1.

## 3.1 Stopping Times

Informally, a stopping time of a stochastic process $\mathbf{x}$ is a random index $\tau$ for which the occurrence of an event $\{\tau = i\}$ can be determined based only on the information observed in the prefix $\mathbf{x}_{1:i}$ (modeled by a sigma algebra $\mathcal{F}_i$). A classic example of a stopping time is a *first hit time*: the first index at which a stochastic process $\mathbf{x}$ attains a value $v$. This is a stopping time because the condition $\{\tau = i\}$ can be determined at time $i$ simply by inspecting whether $\mathbf{x}_i = v$. An example of a random time that is not a stopping time is a *last exit time*: the last index $i$ at which $\mathbf{x}_i = v$. In contrast to the first hit time, the last exit time can only be determined after observing the entire process $\mathbf{x}$.

**Definition 3.3.** (Stopping Times) Let $I$ be an ordered index set, let $(\Omega, \mathcal{F}, (\mathcal{F}_i)_{i \in I})$ be a filtered measurable space, and let $\tau : \Omega \to I$ be a random index defined on this space. We say that $\tau$ is a *stopping time* if $\{\tau = i\} \in \mathcal{F}_i$ for all $i \in I$.

In our case, $I = \{1, \ldots, N + K\}$, and $\tau$ is a random index into the sequence $\mathbf{a}_{1:N+K}$. The filtration $(\mathcal{F}_i)_{i \in I}$ consists of the sigma algebras $\mathcal{F}_i$ generated by the prefix sequences $\mathbf{a}_{1:i}$. While we adopt the conventional terminology of stopping *times*, in this case it might be better to think of $\tau$ as a stopping *index* of $\mathbf{a}_{1:N+K}$, not to be confused with a (continuous) random time in the underlying point process.

To condition on a control $\mathbf{u}_k$, it is essential that $\tau_{\mathbf{u}_k} - 1$ is a stopping time: inference relies on a criterion computed at each index $i - 1$ to determine whether to (temporarily) stop sampling and insert a control $\mathbf{a}_i = \mathbf{u}_k$, or to continue sampling events $\mathbf{a}_i \sim p(\mathbf{e}_i | \mathbf{a}_{1:i-1})$ (see Section 3.4). During inference, we only have access to the prefix $\mathbf{a}_{1:i-1}$ and therefore the criterion that determines whether to condition on $\mathbf{u}_k$ at index $i$ must be a function of this history. Whereas $\tau_{\mathbf{u}_k} - 1$ is a stopping time, examples of random indices that do *not* appear after stopping times include:

1. $\sigma_{\mathbf{u}_k} = k + \arg\min_j \{t_{j+1} \geq s_k - \delta\}$. This is where $\mathbf{u}_k$ would appear in sort order, as if $\mathbf{u}_k$ were at time $s_k - \delta$ (naively anticipating $\mathbf{u}_k$, $\delta$ seconds in advance).

2. $\sigma_{\mathbf{u}_k} = k + \arg\min_j \{t_{j+10} \geq s_k\}$. This is 10 indices before where $\mathbf{u}_k$ would appear in sort order, as if $\mathbf{u}_k$ were an event at time $s_k$ (naively anticipating $\mathbf{u}_k$, 10 indices in advance).

In both cases, $\{\sigma_{\mathbf{u}_k} = i\}$ cannot be determined based on observation of the prefix $\mathbf{a}_{1:i-1}$. Each random index depends on unobserved future events and therefore $\sigma_{\mathbf{u}_k} - 1$ is not a stopping time.

## 3.2 Sparse Sequences

The interleaving rule proposed in Definition 3.1 provides no guarantee that a control will appear some number of seconds (or number of indices) in advance of the time that it controls. For very sparse sequences (relative to $\delta$) a control can even appear *after* the time that it controls.

**Example 3.4.** Suppose $t_1 = 1$, $t_2 = 2$, $t_3 = 5$, $s_1 = 4.5$, and $\delta = 2$. Then $s_1 - \delta = 2.5$ and

$$\tau_{\mathbf{e}_1} = 1 + \arg\max_k \{-\infty \geq s_k - \delta\} = 1, \tag{10}$$

$$\tau_{\mathbf{e}_2} = 2 + \arg\max_k \{1 \geq s_k - \delta\} = 2, \tag{11}$$

$$\tau_{\mathbf{e}_3} = 3 + \arg\max_k \{3 \geq s_k - \delta\} = 3, \tag{12}$$

$$\tau_{\mathbf{u}_1} = 1 + \arg\min_j \{t_j \geq 2.5\} = 4. \tag{13}$$

Therefore $\mathbf{a}_{1:4} = (\mathbf{e}_1, \mathbf{e}_2, \mathbf{e}_3, \mathbf{u}_1)$. And in particular, the control $\mathbf{u}_1$ on time $s_1 = 4.5$ appears after event $\mathbf{e}_3$, which occurs at time $t_3 = 5$.

**Definition 3.5.** Given a sequence of events $\mathbf{e}_{1:N}$ that occur at times $\mathbf{t}_{1:N}$, let $\Delta_{\max}$ denote the maximum distance between adjacent events, i.e.

$$\Delta_{\max} = \max\{\mathbf{t}_{i+1} - \mathbf{t}_i : 1 \leq i < N\}. \tag{14}$$

We say that the sequence $\mathbf{e}_{1:N}$ is $\Delta_{\max}$-*dense*.

For a $\Delta_{\max}$-dense sequence $\mathbf{e}_{1:N}$, we can guarantee that a control on time $\mathbf{t}$ appears at or before time $\mathbf{t} - \delta + \Delta_{\max}$ using the sequence order given by Definition 3.1. If $\delta - \Delta_{\max} > 0$ is small, the model may have little time to plan for anticipated controls; if $\Delta_{\max} > \delta$ then we risk anticipating some controls after the times that they are supposed to control, as in Example 3.4. To ensure dense sequences, we insert special REST events into inter-event intervals that exceed a target density $\Delta^*$: if $n\Delta^* < \mathbf{t}_{i+1} - \mathbf{t}_i \leq (n+1)\Delta^*$ then we insert $n$ REST events at times $\mathbf{t}_i + \Delta^*, \mathbf{t}_i + 2\Delta^*, \ldots, \mathbf{t}_i + n\Delta^*$. This is analogous to how a musician counts out rests in a musical score.

**Example 3.6.** Continuing Example 3.4, we see that $\Delta_{\max} = 3 > 2 = \delta$. Inserting REST events to ensure $\Delta^*$-density for $\Delta^* = 1$, the event sequence becomes $\mathbf{e}'_{1:5}$, adding REST tokens at times 3 and 4 such that

$$\mathbf{e}'_1 = \mathbf{e}_1, \qquad \mathbf{e}'_2 = \mathbf{e}_2, \qquad \mathbf{e}'_3 = (3, \text{REST}), \qquad \mathbf{e}'_4 = (4, \text{REST}), \qquad \mathbf{e}'_5 = \mathbf{e}_3. \tag{15}$$

In this case, the anticipatory interleaving of $\mathbf{u}_1$ with $\mathbf{e}'_{1:5}$ is $\mathbf{a}' = (\mathbf{e}'_1, \mathbf{e}'_2, \mathbf{e}'_3, \mathbf{u}_1, \mathbf{e}'_4, \mathbf{e}'_5)$. We anticipate the control $\mathbf{u}_1$ on time $s_1 = 4.5$ between the event $\mathbf{e}'_3$ at time 3 and the event $\mathbf{e}'_4$ at time 4. All the anticipatory models in this paper are trained with $\Delta^* = 1$ second.

### 3.3 Training Anticipatory Models

We train anticipatory autoregressive models using standard maximum likelihood estimation of the sequences $\mathbf{a}_{1:N+K}$ (Definition 3.1). We tokenize these sequences according to the encoding described in Definition 2.2: $\mathbf{x}_{1:3(N+K)} \equiv \texttt{tokenize}(\mathbf{a}_{1:N+K})$. We follow a standard sequence packing procedure (see, e.g, Appendix B of Brown et al. (2020)) to construct training examples of fixed length $M$ (the model context) from variable-length sequences $\mathbf{x}_{1:3(N+K)}$, using a special event SEP as the sequence separator. We prepend each training example with a single global control code $\mathbf{z} \in \{0, 1\}$ that indicates whether the example contains local controls $\mathbf{u}_k$; setting $\mathbf{z} = 0$ (no controls) facilitates comparisons between anticipatory and autoregressive models. If the training example spans multiple sequences, then $\mathbf{z}$ describes the sequence preceding the first SEP event. We randomly shuffle and mini-batch training examples for stochastic gradient training.

We can choose to either predict all of $\mathbf{a}_{1:N+K}$, learning a *joint* generative model over events and controls, or alternatively just predict the events $\mathbf{e}_{1:N}$ (by zeroing out the training losses at indices corresponding to controls) and learn a *conditional* generative model over the events, given controls. In this paper, we predict controls in addition to the events: this maximizes the number of predictions for each example ($M-1$ predictions for a training example of length $M$). For the infilling application (Section 3.5) predicting events and predicting control are similar enough tasks that improvements in the two tasks ought to reinforce each other; this reinforcement has been observed empirically in the language modeling domain (see Donahue et al. (2020), Appendix C).

Dividing up a dataset into training examples of length $M$ introduces boundary effects. For general autoregressive models, this procedure results in an "early token curse," whereby predictions early in a training example must be made with limited context (Press et al., 2021). For anticipatory autoregressive models, more subtle boundary effects arise. For an anticipation interval $\delta$, controls on first $\delta$ seconds of the training example do not appear in the context: these controls appear at the end of the previous training example. If $\delta$ is large relative to $M$ then the boundary effects become severe: many event predictions will be made without the relevant contextual controls, and vice-versa. This tempers the value of making predictions for all $M-1$ sequence indices, imposing a drag on training efficiency for large values $\delta$ relative to $M$. Like the early token curse, this effect is mitigated with larger contexts $M$ relative to $\delta$; in practice, the maximum practical anticipation interval $\delta$ is thus coupled with the context size of the model that we plan to train.

---

**Algorithm 1:** Anticipatory Autoregressive Sampling

---

**Parameters:** Anticipatory autoregressive model $p$ with context length $M$
Anticipation interval $\delta > 0$
**Input**      : Time-localized controls $\mathbf{u}_{1:K}$ (monotone increasing in time)
Non-localized controls $\mathbf{z}$ (global control codes)
**Output**    : A generated sequence $\mathbf{a}_{1:N+K}$

---

$\mathbf{a}_0 \leftarrow \texttt{SEP}$                                // A special sequence separator event
$i \leftarrow 1$                       // Index $i$ tracks position in the generated sequence
$k \leftarrow 0$                       // Index $k$ tracks position in the control sequence
**do**
    $t \leftarrow \texttt{time}(\mathbf{a}_{i-1})$               // Get the time $t$ of the previous event $\mathbf{a}_{i-1}$
    **while** $\texttt{time}(\mathbf{u}_k) \leq t + \delta$         // While there are controls before time $t + \delta$
    **do**
        $\mathbf{a}_i \leftarrow \mathbf{u}_k$                // Anticipate control $\mathbf{u}_k$ at index $i$
        $i \leftarrow i + 1$                    // Advance to index $i + 1$
        $k \leftarrow k + 1$                   // Consume control $\mathbf{u}_k$
    **end**
    Sample $\mathbf{a}_i \sim p(\cdot | \mathbf{z}, \mathbf{a}_{i-M-\text{Length}(\mathbf{z}):i-1})$     // Sample an event from the model
    $i \leftarrow i + 1$                      // Advance to index $i + 1$
**while** $\mathbf{a}_i \neq \texttt{SEP}$
**return** $\mathbf{a}_{1:i-1}$                        // The value N+K = i-1

---

### 3.4 Anticipatory Inference

We draw conditional samples from an anticipatory autoregressive model $p$ according to the procedure described in Algorithm 1. If there are controls ($K \neq 0$) we set $\mathbf{z} = 1$ (anticipatory sampling mode). In the outer loop, we sample $\mathbf{a}_i \sim p(\cdot | \mathbf{a}_{1:i-1})$. We impose three constraints upon the sampling distribution, all implemented by logit masking: (1) each arrival time must equal or exceed the previous arrival time (monotonicity) (2) a time token must follow a note token, a duration token must follow a time token, and a note token must follow a duration token (proper ordering) and (3) the model must not generate a control token (controls are pre-specified by the user).

Each time we draw a sample $\mathbf{a}_i$ from the model, we note its time $t_i = \texttt{time}(\mathbf{a}_i)$ and check for controls satisfying the condition given by Equation (4) (leveraging the fact that this condition is a stopping time). We anticipate these controls by appending them to the generated output following $\mathbf{a}_i$. The sequence $\mathbf{a}_{1:N+K}$ returned by Algorithm 1 contains $N$ events interleaved with $K$ labels. We can postprocess this sequence by stripping out the controls to recover the sequence of generated events $\mathbf{e}_{1:N}$.

### 3.5 Anticipatory Infilling Models

We apply the anticipatory modeling framework to infilling control, whereby the labels $\mathbf{u}_{1:K}$ consist of a subset of the events $\mathbf{e}_{1:N}$. We duplicate the event vocabulary to distinguish between regular events $\mathbf{e}'_j \in \mathcal{V}$ and control events $\mathbf{u}_k \in \mathcal{U}$; let $\varphi : \mathcal{V} \rightarrow \mathcal{U}$ denote the (bijective) map between the event and control vocabularies. In this case, the combined sequence $\mathbf{a}_{1:(N-K)+K} = \mathbf{a}_{1:N}$ (Definition 3.1) is a re-ordering of the event sequence $\mathbf{e}_{1:N}$, consisting of events $\mathbf{e}'_{1:N-K} \subseteq \mathbf{e}_{1:N}$ and controls $\mathbf{u}_{1:K}$, with $\varphi^{-1}(\mathbf{u}_{1:K}) \subseteq \mathbf{e}_{1:N}$ and $\mathbf{e}_{1:N-K} \cap \varphi^{-1}(\mathbf{u}_{1:K}) = \varnothing$. Using an arrival-time tokenization of events (Definition 2.2) allows us to re-order $\mathbf{e}_{1:N}$ while preserving the semantics of the sequence. We can recover the original sequence by converting the control events back to the event vocabulary and sorting all the events according to their arrival times:

$$\mathbf{e}'_{1:N-K}, \mathbf{u}_{1:K} = \texttt{split}(\mathbf{a}_{1:N}), \tag{16}$$

$$\mathbf{e}_{1:N} = \texttt{sort}\left(\mathbf{e}'_{1:N-K} \cup \varphi^{-1}(\mathbf{u}_{1:K})\right). \tag{17}$$

We describe the precise encoding of $\mathcal{U} \cup \mathcal{V}$ that we use for music infilling models in Appendix C.

Previously we assumed that the controls $\mathbf{u}_{1:K}$ are given to us, separate from the the events $\mathbf{e}_{1:N}$. For the infilling task, there is no distinction between events and controls in the training data; a user could specify an arbitrary set of events to condition on at inference time. To train an infilling model, we must impose a prior on the distribution over subsets of control events $\mathbf{u}_{1:K} \subseteq \mathbf{e}_{1:N}$. This prior should simulate common infilling patterns, and generalize to accommodate patterns we did not foresee during training. To that end, we propose a distribution over control events consisting of a mixture of random spans of time, random subgroups of instrument, and uniformly random events. We describe specifics of this prior in Appendix D.

We caution that the conditionals learned by an anticipatory infilling model are not consistent with a unique joint distribution over sequences $\mathbf{e}_{1:N}$. Given controls $\mathbf{u}_{1:K}$, the map between the sequences $\mathbf{e}_{1:N}$ and $\mathbf{a}_{1:N}$ defined by Equation (17) is a bijection. The probability distribution over sequences $\mathbf{a}_{1:N}$ therefore also defines a distribution over sequences $\mathbf{e}_{1:N}$. Every subset $\varphi^{-1}(\mathbf{u}_{1:K}) \subseteq \mathbf{e}_{1:N}$ can be used as infilling controls, each resulting in a distinct anticipatory sequence $\mathbf{a}_{1:N}$. But the probability distributions $p(\mathbf{e}_{1:N})$ implied by sequences $\mathbf{a}_{1:N}$ will not be the same. We rely on the model's ability to well-approximate the data distribution to ensure approximate consistency between these families of learned distributions.

## 4 Anticipatory Infilling Models of Music

We train and release anticipatory infilling models on the Lakh MIDI dataset (Raffel, 2016). See Appendix B for licensing information and Section A for a discussion of copyright considerations regarding models trained on Lakh MIDI. The Lakh MIDI dataset consists of 178,561 MIDI files (event sequences) that we preprocess into 663,555,310 events (1,990,665,930 tokens using arrival-time encoding) encompassing 8943 hours of music. This dataset is orders of magnitude larger than other common music datasets (Dong et al., 2020), but orders of magnitude smaller than the datasets used to train large models in other domains, e.g. language (Gao et al., 2020) and vision (Schuhmann et al., 2022). For reference, the OpenWebText corpus that reproduces the training set used to train GPT-2 contains approximately 10 billion tokens (Gokaslan & Cohen, 2019). We describe training splits and additional information about Lakh MIDI and preprocessing in Appendix E.

All models trained in this paper are parameterized using causal masked transformers (Vaswani et al., 2017) (decoder-only models) with a context length of $M = 1024$ tokens (defined in Section 3.3). We train models at three scales, following GPT-2 naming conventions (Radford et al., 2019): Small (128M parameters), Medium (360M parameters), and Large (780M parameters). Because anticipatory models are trained like sequence models on the augmented sequences $\mathbf{a}_{1:N+K}$ (see Section 3.3) we are able to use standard libraries for training anticipatory music transformers; in this work, we use the Levanter library for training.[2] For additional details of models and training procedures, see Appendix G.

Anticipatory models are trained with an anticipation interval of $\delta = 5$ seconds; this allows the models to look ahead (i.e. anticipate) controls up to 5 seconds before their arrival. This interval is chosen to maximize anticipation, while not anticipating controls so far in advance that a control on time $s$ might appear more than $M$ tokens earlier than events near time $s$ (outside of the model context $M$). Music in the Lakh MIDI dataset is represented using an average of 68 tokens per second, with a standard deviation of 51 tokens per second (see Appendix E). Therefore, setting $\delta = 5$ seconds ensures that controls on time $s$ appear within $M = 1024$ tokens of events near time $s$, unless tokens are generated at a sustained rate of more than 2.67 standard deviations above the mean (i.e., more than $204 \approx 68 + 2.67 * 51$ tokens/second $\approx M/\delta$).

### 4.1 Automatic Metrics

See Table 1 for a summary of configurations and performance metrics derived from the log-loss for anticipatory infilling models trained on Lakh MIDI. Arrival-time encoding appears more effective for training autoregressive transformers than interarrival-time encoding (rows 1 and 2). Anticipatory training incurs a small tax on performance compared to a baseline autoregressive model for short training schedules (compare rows 2 and 3) however this gap disappears with a longer training schedule (compare rows 4 and 5). In this sense, anticipation unlocks infilling control capabilities in autoregressive music models "for free." Larger models achieve significantly better perplexity (compare rows 3, 6, and 9) as do longer training schedules (compare rows 2 and 4; 3 and 5; 6, 7, and 8).

---

[2]https://github.com/stanford-crfm/levanter

Table 1: Evaluation Loss. All losses are reported on the Lakh MIDI test set sequences $\mathbf{e}_{1:N}$ (without anticipation). For the arrival-time models, we define a per-event loss $L_\mathbf{e}$ summed across event triples (Definition 2.2) and report the per-event perplexity $\mathrm{ppl}(\mathbf{e}) = \exp(L_\mathbf{e})$. This loss decomposes into $L_\mathbf{e} = L_\mathbf{t} + L_\mathbf{d} + L_\mathbf{n}$ with corresponding perplexities for onsets $\mathbf{t}$, durations $\mathbf{d}$ and notes $\mathbf{n}$. The time-normalized bits per second metric (bps) (Thickstun et al., 2019) is defined in Appendix F. Parameter counts differ between arrival and interarrival models, due to the difference in vocabulary size. Anticipatory models are indicated by the 'AM' flag. For anticipatory models, we evaluate with $\mathbf{z} = 0$ (see Section 3.3).

| # | Config | Params | Steps | Encoding | AM | bps | ppl($\mathbf{e}$) | ppl($\mathbf{t}$) | ppl($\mathbf{d}$) | ppl($\mathbf{n}$) |
|---|--------|--------|-------|----------|-----|------|------|------|------|------|
| 1 | Small | 112M | 100k | interarrival | ✗ | 85.9 | - | - | - | - |
| 2 | Small | 128M | 100k | arrival | ✗ | 80.4 | 14.9 | 1.59 | 3.90 | 2.40 |
| 3 | Small | 128M | 100k | arrival | ✓ | 80.7 | 15.0 | 1.58 | 3.98 | 2.40 |
| 4 | Small | 128M | 800k | arrival | ✗ | 75.7 | 12.7 | 1.53 | 3.65 | 2.27 |
| 5 | Small | 128M | 800k | arrival | ✓ | 75.0 | 12.4 | 1.52 | 3.64 | 2.24 |
| 6 | Medium | 360M | 100k | arrival | ✓ | 74.4 | 12.1 | 1.54 | 3.55 | 2.22 |
| 7 | Medium | 360M | 200k | arrival | ✓ | 71.5 | 11.1 | 1.51 | 3.39 | 2.16 |
| 8 | Medium | 360M | 800k | arrival | ✓ | 69.7 | 10.4 | 1.49 | 3.29 | 2.12 |
| 9 | Large | 780M | 100k | arrival | ✓ | 73.2 | 11.7 | 1.52 | 3.44 | 2.23 |

## 4.2 Human Evaluation

We solicited human evaluation of generated music to ground the performance of these models in human assessments of quality, to compare these models to other music generation systems, and to evaluate the anticipatory capabilities of these models that are not captured by the predictive log-loss. We evaluate generated outputs for two tasks—described below—following a similar procedure to Huang et al. (2018), whereby we ask workers to identify which of two 20-second synthesized audio clips is more *conventionally musical*. We recruited crowd workers on the Amazon Mechanical Turk platform to perform these tasks. We paid workers $0.75 US dollars for each pairwise evaluation. Assuming that workers listen to each clip twice—and spend an additional 40 seconds to make their decision and overhead time between tasks—this amounts to two minutes of time per task, or a $22.50 hourly rate. We pre-qualified workers for by asking them to distinguish between five pairs of human compositions versus melodies accompanied by the random retrieval baseline (described below for the accompaniment task).

All samples from our models used for human evaluation are generated using nucleus sampling (Holtzman et al., 2020) with top-p probability $p = 0.95$. We chose this threshold by manually inspecting music generated from the 800k-step Medium model (Row 8 in Table 1) conditioned on prompts from the validation set and using values of $p \in \{0.9, 0.95, 0.98, 1.0\}$. We chose the value $p$ that most consistently produced high quality music (in the authors' judgement). Sampling with $p < 1.0$ generates less diverse music; anecdotally we observe that music generated at $p = 0.95$ is more conservative than both music generated at $p = 1.0$ and music composed by humans (e.g., sampling at $p = 0.95$ is less likely to introduce a new instrumental part). It is possible that our instruction for workers to identify the most conventionally musical clip favors this more conservative generated music.

**Prompt Continuation.** Workers evaluated 50 clips, each consisting of a three-bar prompt—randomly selected from the beginning of tracks in the Lakh MIDI test set—followed by a continuation generated by a model or from the original human composition. See Table 2 for human evaluation of music generated from prompts, compared to human compositions; for full pairwise comparison results, see Appendix J. We use this task to compare our model in the standard autoregressive setting to a baseline Music Transformer (Huang et al., 2018) as implemented by von Rütte et al. (2023) and trained on the Lakh MIDI dataset; we refer to this as the FIGARO baseline. While continuing a prompt does not require anticipation, evaluators find that music generated by anticipatory models given a prompt is considerably more musical than music generated by the baseline FIGARO model (this should not be surprising; the FIGARO Music Transformer is a smaller, 30M parameter model). Human evaluators prefer the Small arrival-time model (Row 3; Table 1) over the Small interarrival-time model (Row 1; Table 1) consistent with the superior log-loss of the arrival-time model.

Table 2: Human evaluation of generated continuations of three-bar musical prompts versus human compositions. P-values are reported using a Wilcoxon signed rank test. Row numbers reference Table 1.

| Model | Wins | Ties | Losses | p-value |
|---|---|---|---|---|
| Medium (Row 8) | 44 | 29 | 77 | 0.0027 |
| Music Transformer (von Rütte et al., 2023) | 13 | 18 | 119 | $2.806 \times 10^{-20}$ |
| Small (Row 3) | 43 | 23 | 84 | 0.0002 |
| Small (Row 1) | 31 | 16 | 103 | $4.976 \times 10^{-10}$ |

Table 3: Human evaluation of 15-second accompaniments versus human compositions. P-values are reported using a Wilcoxon signed rank test.

| Algorithm B | Wins | Ties | Losses | p-value |
|---|---|---|---|---|
| Anticipatory | 18 | 31 | 11 | 0.194 |
| Autoregressive | 5 | 10 | 45 | $1.542 \times 10^{-08}$ |
| Retrieval | 2 | 6 | 52 | $1.017 \times 10^{-11}$ |

**Accompaniment.** Workers evaluated 20 clips, each consisting of a five-second prompt (randomly selected from tracks in the Lakh MIDI test set) followed by a 15-second *accompaniment* (i.e. infilling) conditioned on the prompt and the full (20 seconds) melodic line. See Table 3 for human evaluation of generated music accompaniments, compared to human compositions; for full pairwise comparison results, see Appendix J. For this task, we crudely define *melodic line* to be the instrumental part with the highest pitch. This task allows us to probe the infilling capabilities of our anticipatory autoregressive models. We compare anticipation to two baseline accompaniment algorithms: random retrieval and autoregressive accompaniment. Random retrieval is a simple baseline whereby we accompany the melody with a random 15-second clip retrieved from elsewhere in the track. Autoregressive accompaniment is an algorithmic attempt to use autoregressive generation (without anticipation) to solve the accompaniment task; see Appendix I for details. Both anticipatory and autoregressive accompaniments are generated using a Medium anticipatory model (Row 8; Table 1). Evaluators express a mild preference for anticipatory accompaniments over the human composition skyline. While this result is not statistically significant, it points to the effectiveness of anticipation, especially in the more constrained accompaniment setting.

## 5    Related Work

**Controllable Generative Modeling.** Anticipatory infilling models are motivated in part by control capabilities of recent text infilling models (Zhu et al., 2019; Du et al., 2022; Aghajanyan et al., 2022) and the growing empirical evidence that autoregressive models can be augmented with infilling objectives without sacrificing unconditional modeling performance (Donahue et al., 2020; Bavarian et al., 2022). An analogous approach to span-infilling for music using Seq2Seq conditioning was proposed by Ippolito et al. (2018). Our discussion of anticipation draws inspiration from the asynchronous control setting considered by Hassibi et al. (1999). Anticipatory infilling models also draw inspiration from orderless NADE (Uria et al., 2014) and XLNet (Yang et al., 2019), which learn ensembles of models over different autoregressive factorizations. Anticipatory infilling models differ from these methods by (1) applying *local* rather than global permutations of the factorization order and (2) achieving this by permuting the sequence itself (facilitated by context-free arrival-time encoding; Section 2.1) rather than masking it.

Markov-chain Monte Carlo (MCMC) samplers are a natural candidate for conditional sampling tasks, and in particular infilling. Diffusion models have proven highly effective as generative models of images with infilling capabilities (Ho et al., 2020; Lugmayr et al., 2022). Masked language models can also be repurposed as MCMC samplers for infilling (Goyal et al., 2021; Mireshghallah et al., 2022; Wang & Xia, 2021). These models are not easily applied to conditional point process generation, which requires the generation of variable numbers of events within a specified region of space or time; for work developing MCMC infilling methods in restricted families of point processes, see Shelton et al. (2018). Applications of MCMC methods for symbolic

music infilling have compromised by either discretizing time with piano-rolls (Hadjeres et al., 2017; Huang et al., 2017) or modeling coarse, fixed-rate latent variables, without the ability to control granular details of the variable-length sequence (Mittal et al., 2021).

Sequential Monte Carlo (SMC) samplers have been considered for infilling of both continuous temporal point processes (Mei et al., 2019) and sequence models (Lin & Eisner, 2018); the later more directly relates to our setting because discretizing time (Section 2.1) reduces point process modeling to sequence modeling. These SMC methods approximate samples from the conditional distribution over events given all past and future controls, using importance sampling to select among candidate samples offered by a proposal model. In contrast to SMC, anticipatory inference is more efficient, requiring just one model call to generate an event. The limitations of anticipation are also more explicit: conditional independence of controls more than $\delta$ seconds in the future. In contrast, the limitations of SMC approximations are implicitly determined by the number of candidates and the effectiveness of the proposal model.

**Musical Accompaniment.** In contrast to harmonization tasks, which seek to accompany a melody with simple chords (Simon et al., 2008; Yeh et al., 2021; Chen et al., 2021; Wu et al., 2024; Rhyu et al., 2022), here we seek to generate complete, asynchronous musical accompaniments to a given melody. Huang et al. (2018) previously considered a solo-piano accompaniment generation task using Seq2Seq models; anticipatory models generalize the Seq2Seq approach, motivated by the long multi-instrument sequences found in the Lakh MIDI dataset. Dong et al. (2018) considered an accompaniment task using piano-roll encodings of the Lakh MIDI dataset; see Section 2 for a discussion of the limitations of piano-roll encodings. Both Zhu et al. (2018) and Ren et al. (2020) propose encoder-decoder architectures for conditional generation of accompaniments given a melody: we discuss obstructions to efficiently training encoder-decoder models of music at scale in Section 1. Shih et al. (2022) consider a more abstract form of conditioning, using a melody as thematic material to generate an arrangement—rather than a literal accompaniment—of the given melody. Our accompaniment task is also loosely analogous to recent work on vocal accompaniment in the audio domain (Donahue et al., 2023).

## 6   Conclusion

While the focus of this work has been the music domain, the anticipatory modeling techniques developed in Section 3 are generally applicable to the controllable generation of temporal point processes. Temporal point processes appear in any setting where data is associated with timestamps, including ecommerce activity (Du et al., 2015), social media logs (Farajtabar et al., 2017), healthcare records (Enguehard et al., 2020), and neuroscience data (Williams et al., 2020). Using analogs to the encoding developed for music in Section 2, anticipatory modeling could be used to facilitate the controllable simulation of data in these and other modalities. Furthermore, while the experiments in this paper focus on infilling control–a special case where the controls and events belong to the same modality–anticipatory modeling could be applied more generally to simulate events in one modality, guided asynchronous by controls in a different modality.

Provided that locality continues to be an important inductive bias for learning, we believe that anticipation will be useful for modeling conditional temporal point processes. Locality is a perennial theme in the machine learning literature. Popular model architectures including ConvNets (Forsyth et al., 1999) and LSTM (Hochreiter & Schmidhuber, 1997) exploit locality as an inductive bias. Many long context Transformers adopt local sparsity as an approximation to dense attention (Child et al., 2019; Beltagy et al., 2020; Zaheer et al., 2020). Locality can be exploited to improve training efficiency via staged training (Press et al., 2021; 2022). The local structure of anticipatory sequences is conducive to the broad class of methods that exploit or depend upon locality. Because anticipation only intervenes to modify the data, it can be seamlessly combined with other modeling innovations, e.g., in the music space: the relative transformer (Huang et al., 2018), the compound word transformer (Hsiao et al., 2021), and RIPO attention (Guo et al., 2022).

Generative music models have not yet reached a broad audience alongside generative models of language (Brown et al., 2020) or images (Ramesh et al., 2022). Slow adoption of music models within creative communities is partly attributable to the difficulty of controlling these models (Briot & Pachet, 2020). Users value agency in human-AI collaborations, preferring to take an active role over more automated solutions (Oh

et al., 2018; Roy et al., 2019). We note the proliferation of recent work on music generation (primarily in the audio domain) controlled by text (Forsgren & Martiros, 2022; Kreuk et al., 2023; Agostinelli et al., 2023; Schneider et al., 2023; Huang et al., 2023a;b); we are intrigued by the possibility of applying anticipation to generate symbolic music conditioned on localized text labels (e.g., lyrics). More broadly, we view symbolic music generation and audio music synthesis as complementary, analogous to text generation and speech synthesis. We hope that the Anticipatory Music Transformer may fill a role that is currently underdeveloped in the construction of controllable music generation systems that support the human creative process.

## Acknowledgements

We thank Jennifer Brennan, Niladri Chatterji, Peter Henderson, Cheng-Zhi Anna Huang, Sidd Karamcheti, Mina Lee, Mark A. Lemley, Nelson Liu, Michael C. Mozer, Joon Sung Park, Ofir Press, and Dimitri von Rütte for providing invaluable advice, discussion, and support for various aspects of this project. We also thank the crowdworkers who provided impartial human evaluation of the music generated by our models, including: Oliver Crangle, Dare, Razr-Dylan, Bryan Haskins, and Tammy.

This work was done at the Center for Research on Foundation Models (CRFM) at Stanford University. We thank the CRFM and the Stanford Institute for Human-Centered Artificial Intelligence (HAI) for supporting this work. Toyota Research Institute (TRI) and Schmidt Futures provided funds to support this work. The experiments discussed in this paper were conducted on Cloud TPU VMs, provided by the Google TPU Research Cloud (TRC).

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

# A  Ethical Considerations for Generative Models of Music

We find it likely that generative models of music will become widely deployed and used within the next several years (if not sooner). We are particularly concerned about the economic implications of this technology: how will these models affect labor markets for creative work (Section A.1)? We are uncertain about the legal status of these models: how will music created using models be treated by intellectual property and copyright law (Section A.2)? We are also curious how the advent of these models will change our perceptions and experience of art and creativity. Will widespread deployment of models trained on Western music reinforce existing Western hegemony in music culture (Section A.3)? Could the use of these models lead to creative stagnation, or loss of interest in music as an art form (Section A.4)? Analogous questions and concerns have been raised in the broader context of research on generative models, but the focus of this discussion is often grounded in generation of language or images. These discussions are worth revisiting in the specific context of music generation.

## A.1  The Creative Economy and Labor Markets

Music composition has economic value. What happens to the music economy—and to the people who rely on it for their livelihoods—if its creative product can be automatically generated by a machine? Economic research suggests that new technologies exert two countervailing effects on the labor market: a *displacement effect* whereby automation reduces demand for labor, and a *productivity effect* whereby greater productivity increases demand for labor (Acemoglu & Restrepo, 2018). It is unclear to us what the net effect of these two forces will be on the size of the labor market for music composition. If latent demand for new music is high, then decreasing the cost of music production could lead to more employment opportunities for composers. That said, this reconfiguration of the music economy might be disruptive, and success in the new music economy could require a new skillset that is difficult (or unappealing) for current workers to acquire. We suspect that generative music technology will affect the current music economy, and may displace some workers.

One response to concerns about labor displacement is that researchers should focus on building productivity-enhancing systems that augment human capabilities, rather than automating them. This is an appealing perspective for researchers, because it suggests that we can actively steer the development of generative technologies towards pro-social outcomes. Indeed, the methods developed in this paper facilitate the creation of controllable generative models that support the human creative process, rather than automating it. That said, augmentative and productivity-enhancing technologies can still cause labor displacement. A system that augments human capabilities can be used to replace a high-paid, high-skill worker with a low-paid, low-skill worker; or to replace a team of workers with a single highly-productive worker. Augmentation can also change the nature of creative work, potentially devaluing or automating aspects of the job that bring a worker joy.

## A.2  Copyright

While the use of copyrighted data to train models is often defensible under fair use doctrine (2d Circuit, 2015), the use of such data for *generative* modeling is an open legal question (Sobel, 2017; Henderson et al., 2023). To evaluate the copyright status of generative music models, it may be helpful to distinguish between the model itself (i.e. the parameters) and outputs generated by the model (McCann, 2021). It is unclear how the law will ultimately view (the parameters of) a generative model. Is the model a transformative fair use of the training data? Is the model itself a creative construct subject to copyright?

Regarding the outputs of a generative music model: generative models have been shown to plagiarize training data in both the language (Carlini et al., 2023) and vision (Somepalli et al., 2023) domains. Based on this evidence, we are certain that generative models of music are similarly capable of plagiarism. Technical mitigation strategies to prevent this behavior based on differential privacy or near access-freeness are nascent (Vyas et al., 2023), as are socio-technical strategies, such as providing copyright holders an opt-out mechanism to exclude their content from training datasets (Henderson et al., 2023). The models trained in this paper provide no technical guarantees against plagiarism. Therefore, any outputs of this model should be presumed to risk infringing on copyrighted material.

The Lakh MIDI dataset used to train models for this paper is licensed under the Creative Commons CC-BY 4.0 terms, which ostensibly allows free redistribution, reuse, remixing, and transformation of its content for any purpose. However, the contents of this dataset consists of MIDI files aggregated from a variety of sources and subject to their own (heterogeneous, mostly undocumented) copyright terms. In many cases, these MIDI files are derivative work, transcribed from source material (e.g., pop music) which is itself subject to copyright. Therefore, we presume that the copyright status of Lakh MIDI is more restrictive than its license would suggest.

### A.3  Western Bias

While the methods developed in this paper are broadly applicable to modeling diverse musical traditions, the MIDI data format and the Lakh MIDI dataset used to train the anticipatory music transformer both impose a strong bias towards the Western music tradition. In Section 2 we adopted a musical vocabulary that describes musical pitch according to the 12-tone chromatic scale, which we inherit from the MIDI format of our training data. Models trained using this vocabulary are incapable of expressing music outside the 12-tone scale. This excludes, for example, the Koron and Sori quarter tones of Persian Dastgah music (Abdoli, 2011), the 53 Koma of Turkish makam (Benetos & Holzapfel, 2013), and the continuous-pitch gamaka appearing in Carnatic music (Viraraghavan et al., 2018). For further discussion of the consequences of modeling using a 12-tone discrete scale, see Lenchitz (2021).

The Lakh MIDI dataset used to train anticipatory music transformers in Section 4 consists almost exclusively of Western music. This inevitably biases the model's predictions towards infilling completions that follow Western rules of composition. We encourage work towards building models for other music traditions. We believe that the primary obstruction to this work will be the limited availability of data. In this sense, extending the work presented in this paper to other music traditions may be analogous to work in the natural language processing community on low-resource language modeling (Hedderich et al., 2021).

### A.4  Models and Art

If generated music becomes too good, might people lose interest in creating music altogether? We find this unlikely. Our appreciation and emotional attachment to art is closely connected to its provenance, the story behind its creation, and the story of its creator. Machine-generated music is unlikely to supplant high art, although we expect that many artists may incorporate generative tools into their creative processes. Economic pressures aside, artists are free to use generative tools or not. We expect many artists to embrace these tools, but surely others will eschew them in favor of more traditional creative processes.

From a humanist perspective, we are optimistic for a future with generative models. Due to the high skill required to compose music, most people are currently unable to create music at all. Lowering the barrier to music creation is more likely to increase interest in music than to decrease it. A strong generative model of music can be repurposed as a learning tool, providing feedback on compositions and ideas for how to improve a composition (Huang et al., 2016). We draw an analogy to modern chess engines, which play the game of chess at a far higher level than even the best human players. Despite this, public interest in the game of chess has never been higher (CHESScom, 2023): we celebrate human performance, and it is irrelevant that the computer on our phone could beat us every time. Chess engines offer universal access to a form of chess education: whereas chess development previously required tutoring by a strong chess player (requiring money and access) young players today can learn to play well by inspecting how the engine plays (Levinson, 2011). We anticipate similar opportunities for generative models to increase access to music education.

### A.5 Releasing the Anticipatory Music Transformer

We see many opportunities for generative models to support human creativity and expression. At the same time, we are concerned for the economic prospects of current participants in the music labor market, whose work may disrupted or displaced by the deployment of generative music technologies. In our work, we strive to develop methods that maximize the humanistic potential for generative models of music. Specifically, we focus on controllable generation, which places the expressive power of these models under the control of their users. We focus on generation of intermediate *symbolic* music representations, which could be integrated as an assistive tool—analogous to a writing assistant (Lee et al., 2022)—into the music sequencing and synthesis workflow of a modern digital audio workstation. In contrast to audio models, symbolic music generation could provide a less disruptive integration of generative music technologies into existing workflows and thus support current participants in the music labor market. Nevertheless, we find it impossible to weigh the opportunities presented by these models against the challenges this technology might pose for workers in the music economy. Criticisms of capitalist economic structures—that accrue the windfalls of automation and productivity-enhancing technologies neither to their inventors nor to workers whose labor they displace—are beyond the scope of this paper. We welcome feedback on our decision to pursue this line of research; we hope to foster a discussion of how we can steer future research in this area towards methods that serve and support composers and musicians.

## B   Licensing

We release the code for constructing anticipatory infilling models and weights for the models discussed in this paper under the Apache License, Version 2.0.

## C   Encoding Details for Anticipatory Infilling Models.

The tokenization of training examples (Section 3.3) for anticipatory infilling models using an arrival-time encoding of events (Definition 2.2) is described by Definition C.1. As discussed in Section 3.5, we double the base vocabulary to distinguish between anticipated events and non-anticipated events. We also include a sequence separator token and global control codes $\mathbf{z} \in \{\texttt{AR}, \texttt{AAR}\}$ (Section 3.3) as well as the REST token (Section 3.2).

**Definition C.1.** (Arrival-Time Training Example) Let $\mathbf{a}_{1:(M-1)/3}$ be a training example, possibly re-ordered according to Definition 3.1 for infilling. An *arrival-time tokenized training example* is a sequence $\mathbf{x}_{1:M}$, defined as follows. We define two special tokens REST $= 27512$ and SEP $= 55025$, and two control tokens AR $= 55026$ and AAR $= 55027$. Separation between two sequences is indicated by a triple of three SEP tokens. The first token $\mathbf{x}_1$ in every training example encodes the global control code $\mathbf{z}$:

$$\mathbf{x}_1 \in \{\texttt{AR}, \texttt{AAR}\}, \qquad \text{(the anticipation control code).} \quad (18)$$

If $\mathbf{a}_i$ is a non-anticipated event then

$$\mathbf{x}_{3i-2} \in \{0, \dots, 10000\} \cup \{\texttt{SEP}\}, \qquad (0-100\text{s}, 10\text{ms quantized}) \quad (19)$$
$$\mathbf{x}_{3i-1} \in \{10000, 10000+1000\} \cup \{\texttt{SEP}\}, \qquad (0-10\text{s}, 10\text{ms quantized}) \quad (20)$$
$$\mathbf{x}_{3i} \in \{11000, \dots, 11000+16512\} \cup \{\texttt{SEP}\} \cup \{\texttt{REST}\}, \qquad (\text{instruments} \times \text{pitches}). \quad (21)$$

Otherwise (if $\mathbf{a}_i$ is an anticipated event)

$$\mathbf{x}_{3i-2} \in \{27513, \dots, 27513+10000\} \cup \{\texttt{SEP}\}, \qquad (0-100\text{s}, 10\text{ms quantized}) \quad (22)$$
$$\mathbf{x}_{3i-1} \in \{37513, 37513+1000\} \cup \{\texttt{SEP}\}, \qquad (0-10\text{s}, 10\text{ms quantized}) \quad (23)$$
$$\mathbf{x}_{3i} \in \{38513, \dots, 38513+16512\} \cup \{\texttt{SEP}\} \cup \{\texttt{REST}\}, \qquad (\text{instruments} \times \text{pitches}). \quad (24)$$

The total vocabulary size is 55028.

Table 4: Arrival-time tokenization of "Twinkle, Twinkle, Little Star," played on a piano at tempo quarter=120. For clarity, we group the sequence of tokens tokens into triplets (one event per row).

| Token Values | | | Event Description | | | |
|---|---|---|---|---|---|---|
| Arrival Time | Duration | Note | **t** | **d** | **p** | **k** |
| 55,025 | 55,025 | 55,025 | Sequence Separator | | | Event |
| 0 | 10,048 | 11,060 | 0s | 480ms | C4 | piano |
| 50 | 10,048 | 11,060 | 0.5s | 480ms | C4 | piano |
| 100 | 10,048 | 11,067 | 1s | 480ms | G4 | piano |
| 150 | 10,048 | 11,067 | 1.5s | 480ms | G4 | piano |
| 200 | 10,048 | 11,069 | 2s | 480ms | A4 | piano |
| 250 | 10,048 | 11,069 | 2.5s | 480ms | A4 | piano |
| 300 | 10,095 | 11,067 | 3.5s | 950ms | G4 | piano |
| 400 | 10,048 | 11,065 | 4s | 480ms | F4 | piano |
| 450 | 10,048 | 11,065 | 4.5s | 480ms | F4 | piano |
| 500 | 10,048 | 11,064 | 5s | 480ms | E4 | piano |
| 550 | 10,048 | 11,064 | 5.5s | 480ms | E4 | piano |
| 600 | 10,048 | 11,062 | 6s | 480ms | D4 | piano |
| 650 | 10,048 | 11,062 | 6.5s | 480ms | D4 | piano |
| 700 | 10,095 | 11,060 | 7s | 950ms | C4 | piano |

**Example C.2.** The arrival-time tokenization (processed for autoregressive training) of the first for bars of the lullaby "Twinkle, Twinkle, Little Star," played on a piano at tempo quarter=120:

$$\mathbf{x}_{0:46} = [55026, 55025, 55025, 55025, 0, 10048, 11060, 50, 10048, 11060, 100, 10048, 11067,$$
$$150, 10048, 11067, 200, 10048, 11069, 250, 10048, 11069, 300, 10095, 11067, 400, 10048,$$
$$11065, 450, 10048, 11065, 500, 10048, 11064, 550, 10048, 11064, 600, 10048, 11062, 650,$$
$$10048, 11062, 700, 10095, 11060].$$

See Table 4 for a structured description of this sequence.

The tokenization of training sequences for autoregressive models using an interarrival-time encoding (Defintion 2.3) is described by Definition C.3. We double the base vocabulary of notes and instruments to distinguish between onsets and offsets. We truncate interarrival times longer than 10 seconds (to 10 seconds).

**Definition C.3.** (Interarrival-Time Training Example) Let $\mathbf{e}_{1:(M/4)}$ be a training example of events (Definition 2.1). An *interarrival-time tokenized training example* is a sequence $\mathbf{x}_{1:M}$, defined as follows. We define a single special token $\texttt{SEP} = 34024$. Separation between two sequences is indicated by a single $\texttt{SEP}$ token. Using the notation $\mathbf{x}'_{1:(M/2)}$ from Definition 2.3, if $\mathbf{x}'_i$ is an onset,

$$\mathbf{x}_{2i} \in \{1000, \dots, 1000 + 16512\}, \qquad \text{(instruments} \times \text{pitches). (25)}$$

If $\mathbf{x}'_i$ is an offset,

$$\mathbf{x}_{2i} \in \{17512, \dots, 17512 + 16512\}, \qquad \text{(instruments} \times \text{pitches). (26)}$$

And regardless,

$$\mathbf{x}_{2i+1} \in \{0, 1000\}, \qquad \text{(0} - 10\text{s, 10ms quantized). (27)}$$

As described in Definition 2.3, we omit tokens $\mathbf{x}_{2i+1} = 0$. The total vocabulary size is 34025.

**Example C.4.** The interarrival-time tokenization of the first for bars of the lullaby "Twinkle, Twinkle, Little Star," played on a piano at tempo quarter=120:

$\mathbf{x}_{0:56} = [34024, 1060, 48, 17572, 2, 1060, 48, 17572, 2, 1067, 48, 17579, 2, 1067, 48, 17579,$
$2, 1069, 48, 17581, 2, 1069, 48, 17581, 2, 1067, 95, 17579, 5, 1065, 48, 17577, 2, 1065, 48,$
$17577, 2, 1064, 48, 17576, 2, 1064, 48, 17576, 2, 1062, 48, 17574, 2, 1062, 48, 17574, 2, 1060,$
$95, 17572].$

In this case there are no interarrival times of length zero and this tokenization is slightly less compact than arrival-time tokenization. But if we were to extend the length of each note to a full beat, the interarrival-time tokenization becomes:

$\mathbf{x}_{0:43} = [34024, 1060, 50, 17572, 1060, 50, 17572, 1067, 50, 17579, 1067, 50, 17579, 1069, 50,$
$17581, 1069, 50, 17581, 1067, 100, 17579, 1065, 50, 17577, 1065, 50, 17577, 1064, 50, 17576,$
$1064, 50, 17576, 1062, 50, 17574, 1062, 50, 17574, 1060, 100, 17572]$

And in this case, interarrival-time tokenization is slightly more compact.

## D   A Prior over Music Infilling Controls

We propose three types of anticipation, i.e., distributions over events to condition on as controls. First we propose *span anticipation*, whereby we anticipate all tokens in a given span in order to explicitly promote the model's ability to fill-in-the-middle. Second, we propose *instrument anticipation*, whereby we anticipate all tokens except for a specified instrumental part, supporting a workflow whereby supplemental instrumental parts are generated to complement a pre-existing ensemble. And third, we propose *random anticipation*, whereby we randomly anticipate events at some fixed rate to accommodate a broader possible range of user-specified anticipation patterns.

We apply apply these patterns of anticipation to training data according to the following distributions:

- **Span anticipation**. We randomly anticipate consecutive subsequences of events spanning $\delta$ seconds, at an exponential rate $\lambda$. We fix $\lambda = .05$, and at interarrival times $i \sim \text{Exp}(\lambda)$ we anticipate the events $\mathbf{e}_{i:j}$ where $j = \min\{t_j : t_j > t_i + \delta\}$.

- **Instrument anticipation**. For an event sequence with $J$ unique instrument parts, we uniformly sample $j \in \{1, \ldots, J-1\}$ and randomly sample $j$ instrumental parts without replacement. We anticipate all events in the sequence associated with these $j$ parts.

- **Random anticipation**. We uniformly sample a rate $r \in \{0.1, \ldots, 0.9\}$ for each event sequence. We randomly anticipate an $r$ fraction of events in the sequence.

We balance the overall training distribution using the following mix: 10% without anticipation (standard autoregressive training), 10% with span anticipation, 40% with instrument anticipation, and 40% with random anticipation.

These anticipation patterns facilitate interaction with an anticipatory music transformer via user-specified control sequences. For example, the accompaniment task evaluated in Section 4 is modeled by instrument anticipation when $j = 1$. Note that the prior over infilling controls facilitates much more general interaction patterns than the prompt continuation and melodic accompaniment tasks studied in Section 4. We defer further study of the capabilities of anticipatory infilling models to future work.

To train an anticipatory infilling model, we augment the training dataset using the distribution of anticipation patterns specified above. We perform these augmentations during preprocessing, resulting in an augmented dataset derived from the original data. In particular, augment the Lakh MIDI dataset by a factor of 30. Using the prior distribution described above, this augmented dataset contains (i) 3 copies of the original dataset, verbatim; (ii) 3 augmented copies of the dataset with different random anticipated spans; (iii) 12 augmented copies of the dataset with different randomly anticipated instrument subsets for each event sequence; and (iv) 12 augmented copies of the dataset with different randomly anticipated events at different rates for each event sequence.

Table 5: The number of tokens in the Lakh MIDI dataset, using tokenizations described in Section 2.1.

| Encoding | Train | Validation | Test | Overall |
|---|---|---|---|---|
| arrival | 1,741,830,387 | 123,785,046 | 125,050,497 | 1,990,665,930 |
| interarrival | 1,612,129,280 | 114,519,040 | 115,409,920 | 1,842,058,240 |

Augmenting by a factor of 30 results in an arrival-time encoded Lakh MIDI train set of approximately 52B tokens. For training examples of length 1024 and batch size 512, this results in approximately one epoch per 100,000 optimization steps. For models trained with larger step counts, we take multiple passes over the augmented dataset. Given the relatively small size of the Lakh MIDI dataset, anticipation may be of interest as a regularization technique (see Appendix K for some evidence of possible regularizing effects attributable to anticipation). In this case, it could be fruitful to (i) increase the augmentation factor further during training, and (ii) place a smaller weight on training without anticipation, to minimize the amount of duplication of the original (unaugmented) dataset within the augmented dataset.

## E    Details of the Lakh MIDI Dataset

The Lakh MIDI dataset includes no metadata, so we have little fine-grained or quantitative insight into the origins and contents of this dataset. Based on a randomly sampling of the dataset's contents, we observe that it contains many arrangements of modern pop music, transcripts of classical western compositions, and original compositions (with varying degrees of quality).

**Preprocessing**    Out of the 178,561 sequences in the Lakh MIDI dataset, we were able to successfully parse 174,046 sequences.[3] We discard 7032 sequences that are shorter than 100 events or 10 seconds in length. We also discard 40 event sequences that are longer than one hour in length (inspection reveals that many of these sequences are corrupt). Finally, we discard 2227 sequences that contain more than 16 unique instrument parts: representing music with more than 16 parts as MIDI (for synthesizing outputs) requires multiplexing multiple instruments onto MIDI channels; we avoid this complexity by simply excluding very large ensembles. The resulting dataset consists of 164,747 event sequences. We split this dataset in to train, validation, and test splits according to the leading hexadecimal digit of each file's MD5 hash:

- Train: hashes `0`–`d`, 144,202 event sequences, 7827 hours of music.

- Validation: hash `e`, 10,212 event sequences, 555 hours of music.

- Test: hash `f`, 10,333 event sequences, 561 hours of music.

We tokenize the dataset using the arrival-time and interarrival-time encodings described in Section 2.1. As seen in Table 5, the interarrival-time tokenization is slightly more compact. To illustrate the long, variable-rate nature of symbolic music event sequences, we plot the distributions of sequence length (Figure 2) and event rate (Figure 3) for the Lakh MIDI validation split.

## F    Measuring Cross-Entropy in Bits per Second

Because the arrival-time tokenization (Definition 2.2) and the interarrival-time tokenization (Defintion 2.3) describe nearly equivalent information, the log-loss of models trained using either encoding can be meaningfully compared via a unit conversion. To make comparison agnostic to encoding, we report losses in units of bits per second: this is the total log-loss of the test set, divided by the number of seconds of music in the test set. Concretely, given a per-token loss $L$ reported in nats per token, conversions for interrarival time

---

[3]We parse Midi files using the Mido library: https://github.com/mido/mido.

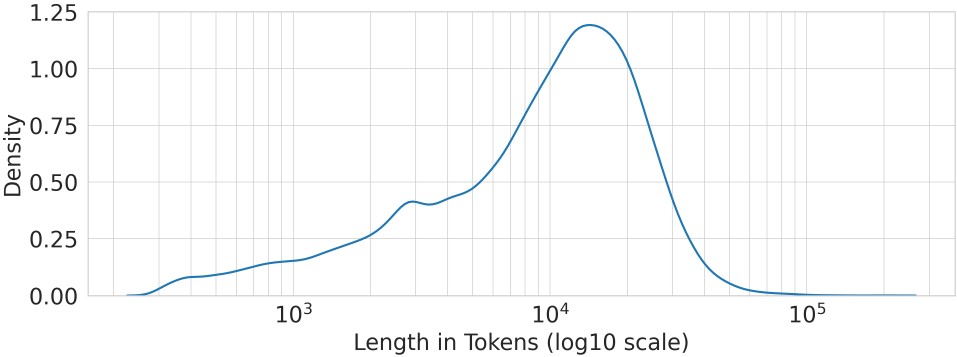

Figure 2: The distribution of sequence lengths calculated for the arrival-time tokenized Lakh MIDI validation split. Mean sequence length is 12,071 tokens, with a standard deviation of 9711 tokens.

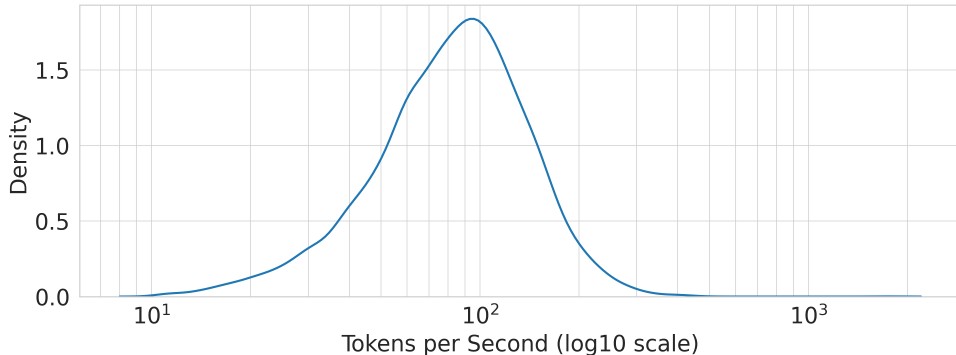

Figure 3: The distribution of instantaneous tokens/second calculated for the arrival-time tokenized Lakh MIDI validation split. Mean instantaneous tokens/second for the Lakh MIDI dataset is 68 with a standard deviation of 51 tokens/second.

and arrival time encodings of the test set defined in Section E are:

$$\text{bps}_{\text{arrival}}(L) = L \times \frac{1}{\log(2)} \times (\texttt{test\_tokens}_{\text{arrival}}/(\texttt{test\_hours} \times 3600)), \tag{28}$$

$$\text{bps}_{\texttt{interarrival}}(L) = L \times \frac{1}{\log(2)} \times (\texttt{test\_tokens}_{\text{interarrival}}/(\texttt{test\_hours} \times 3600)). \tag{29}$$

Concretely, for the Lakh MIDI dataset, from Table 5 we have $\texttt{test\_tokens}_{\text{arrival}} = 125{,}050{,}497$, $\texttt{text\_tokens}_{\text{interarrival}} = 115{,}409{,}920$, and the length of the test set (independent of encoding) is $\texttt{test\_hours} = 560.98$ hours. The remaining factors convert units from nats to bits (the $\log(2)$ factor) and from hours to seconds (the 3600 factor). This normalized form of the log-loss is analogous to the bits per dimension loss commonly reported for image generation (Salimans et al., 2017) and the bits per beat loss for musical scores (Thickstun et al., 2019).

Truncations during tokenization (see Section 2.1) result in slightly different information content between an arrival-time tokenized sequence and the corresponding interarrival-time tokenized sequence. Under arrival-time tokenization, 0.05% of note durations in the Lakh MIDI dataset exceed 10 seconds, and are truncated (to 10 seconds). Under interarrival-time tokenization, a total of 3947 interarrival times (0.00%) in the Lakh MIDI dataset exceed 10 seconds, and are truncated (to 10 seconds). Finally, 0.09% of training examples under arrival-time tokenization exceed 100 seconds in length, and are discarded from the tokenized dataset. These instances are sufficiently rare that they exert a negligible effect on the information content (and therefore on bits per second comparisons) between the arrival-time and interarrival-time tokenized sequences.

Table 6: Model Configurations.

|  | Small | Medium | Large |
|---|---|---|---|
| **Architecture Hyperparameters** | | | |
| Layers | 12 | 24 | 36 |
| Attention Heads | 12 | 16 | 20 |
| Hidden Dimensions | 768 | 1024 | 1280 |
| Sequence Length | | 1024 tokens | |
| Residual Dropout | | 0.1 | |
| Embedding Dropout | | 0.1 | |
| Attention Dropout | | 0.0 | |
| Weight Decay | | 0.1 | |
| **Optimizer Hyperparameters** | | | |
| Max Learning Rate | 0.0006 | 0.0003 | 0.0002 |
| Optimizer | | AdamW (Loshchilov & Hutter, 2018) | |
| $(\beta_1, \beta_2, \epsilon)$ | | $(0.9, 0.999, 1 \times 10^{-8})$ | |
| Batch Size | | 512 sequences (= 524,288 Tokens) | |
| Warmup | | 1000 steps ($\approx$ 50M tokens) | |
| Learning Rate Schedule | | Cosine decay (no restarts) (Loshchilov & Hutter, 2017) | |
| Gradient Clipping | | Clipping above $\|\nabla\| = 1$ (Rae et al., 2021) | |
| **Training Resources** | | | |
| Throughput (tokens/second) | 690,000 | 260,000 | 140,000 |
| Throughput (seconds/iteration) | 0.76 | 2.02 | 3.74 |
| Hardware | | Google TPU v3-32 pod slice | |

## G  Hyperparameters and Resources

All models are parameterized by standard, decoder-only causally masked transformers with GeLU non-linearities (Hendrycks & Gimpel, 2016). The models are implemented in Jax (Bradbury et al., 2018) and trained on Google TPU v3 hardware. We observe that large-scale pseudorandom number generation in Jax is slow, and therefore eschew the standard attention dropout regularization. The models are optimized using AdamW (Loshchilov & Hutter, 2018). The learning rate schedule consists of a 1000 step linear warmup to a maximum learning rate, followed by a single cycle of cosine decay (Loshchilov & Hutter, 2017) over the remaining steps to a final learning rate of zero. Following Chinchilla compute-optimality recommendations, we train each model for a number of steps that is approximately proportional to the model's size (Hoffmann et al., 2022). Configuration details of the models and optimization are presented in Table 6.

Most of these models were trained on TPU v3-32 pod slices, which in practice are approximately equivalent to a GPU machine with 8 NVIDIA A100's. Training throughput for each model configuration using a v3-32 is reported in Table 6. Conversions of these throughput statistics to wall-clock estimates of training time for the models featured in Section 4 are shown in Table 7. Thus, the total training time on v3-32's for the models featured in this paper was approximately

$$1121 \text{ hours} = 3 \times 21 \text{ hours} + 2 \times 169 \text{ hours} + 56 \text{ hours} + 112 \text{ hours} + 448 \text{ hours} + 104 \text{ hours}. \quad (30)$$

In addition to training the models featured in this paper, substantial additional TPU hours were consumed during the development phase of this research. We crudely estimate that total TPU-hours consumed for this work were approximately 3-5 times the hours reported in Equation (30).

Table 7: Estimated wall-clock training time for the Small, Medium, and Large model configurations described in Table 6, using a Google TPU v3-32 pod slice.

| Config | Training Steps | Hours of Training |
|--------|----------------|-------------------|
| Small  | 100k           | 21                |
| Small  | 800k           | 169               |
| Medium | 100k           | 56                |
| Medium | 200k           | 112               |
| Medium | 800k           | 448               |
| Large  | 100k           | 104               |

## H   Details of the FIGARO Music Transformer Baseline

We compare our models to the implementation of Music Transformer (Huang et al., 2018) described by von Rütte et al. (2023). We use the official public implementation for training and sampling from this model.[4] While both our models and this baseline are based on the Transformer architecture and trained on Lakh MIDI, there are at several notable distinctions. The FIGARO Music Transformer architecture implements the relative attention mechanism proposed by (Huang et al., 2018). The FIGARO Music Transformer inputs are tokenized using a REMI encoding that accounts for metrical structure (Huang & Yang, 2020). The FIGARO Music Transformer is also a smaller-scale model than the other models considered in this paper: a six-layer transformer (approximately 30M parameters) trained for 100k iterations on sequences of length 256. Some of these factors (e.g., model scale, sequence length) clearly favor the models presented in this paper; others (e.g., relative attention) may favor the FIGARO model. With so many uncontrolled variables, we caution against drawing conclusions about individual engineering choices in the design of the FIGARO Music Transformer versus the models proposed in this paper.

While pre-trained checkpoints of the FIGARO models are available, the training and evaluation splits used for the pre-trained checkpoints are incompatible with the splits defined in Section E. Therefore, we re-train our own version of the model using hashes 0–d as the training split. A comparison of our re-trained model and the reference model checkpoint are presented in Table 8. For definitions and discussion of the evaluation metrics compared here, see von Rütte et al. (2023). Our version of the FIGARO model matches or slightly outperforms the reference model on these metrics.

Neither our arrival-time tokenization (Definition 2.2) nor FIGARO's REMI encode all the nuances of music described in, e.g., a MIDI file. Because FIGARO explicitly models the metrical structure of music, we prompt using a fixed number of measures (three bars) rather than a fixed amount of time: this is six seconds of prompt material for music in 4/4 time when quarter=120. We select three-bar prompts for the study in the range of 4-6 seconds, thus excluding music with a very fast or slow tempo. To create fair comparisons between music composed by humans, FIGARO, and the Anticipatory Music Transformer, we apply the following procedure for constructing prompts:

- Human compositions: we encode human compositions (initially expressed as MIDI) using the FIGARO tokenizer, and then re-encode these samples using our own tokenizer.

- FIGARO samples: we encode prompts using the FIGARO tokenizer, and re-encode completions of these prompts generated by FIGARO using our own tokenizer.

- Anticipatory Music Transformer: we encode prompts using the FIGARO tokenizer, re-encode the prompts using our own tokenizer, and generate completions from our own models using these re-encoded prompts.

All music is thus restricted to the musical vocabulary of our arrival-time tokenization.

In Table 9, we report the quantitative metrics proposed by von Rütte et al. (2023) using 800 20-second continuations of 3-bar prompts, for both the FIGARO Music Transformer, and our Medium Anticipatory

---

[4]https://github.com/dvruette/figaro

Table 8: Quantitative evaluation metrics for the FIGARO implementation (von Rütte et al., 2023) of Music Transformer (Huang et al., 2018) trained on the FIGARO training data split, compared to the same model trained using the split defined in Section E.

| Train Split | I $\uparrow$ | C $\uparrow$ | TS $\uparrow$ | ND $\downarrow$ | P $\uparrow$ | V $\uparrow$ | D $\uparrow$ | $s_c \uparrow$ | $s_g \uparrow$ |
|---|---|---|---|---|---|---|---|---|---|
| (FIGARO split) | 0.191 | 0.048 | 0.751 | 2.192 | 0.563 | 0.153 | 0.312 | 0.306 | 0.510 |
| Hashes 0—d | 0.207 | 0.050 | 0.770 | 1.523 | 0.564 | 0.158 | 0.289 | 0.305 | 0.517 |

Table 9: Quantitative evaluation metrics for the FIGARO implementation (von Rütte et al., 2023) of Music Transformer (Huang et al., 2018) compared to a Medium Anticipatory Model (Row 8 in Table 1).

| Model | I $\uparrow$ | C $\uparrow$ | ND $\downarrow$ | P $\uparrow$ | D $\uparrow$ | $s_c \uparrow$ | $s_g \uparrow$ |
|---|---|---|---|---|---|---|---|
| von Rütte et al. (2023) | 0.833 | 0.364 | 0.574 | 0.751 | 0.573 | 0.619 | 0.257 |
| Anticipatory | 0.836 | 0.392 | 0.529 | 0.755 | 0.593 | 0.659 | 0.259 |

Music Transformer (Row 8 in Table 1). While the Anticipatory Music Transformer marginally outperforms the FIGARO Music Transformer in all categories, it is not clear that we can draw a strong conclusion from these results. The reported metrics were designed to measure reconstruction quality: how faithfully does a generated output reproduce the original? These metrics make less sense for evaluating open-ended generation: we expect that generated continuations would be different from the originals. One hypothesis for the consistent out-performance of the Anticipatory Music Transformer is that music is often self-similar, and perhaps this model is better able to capture these self-similarities of music.

## I A Baseline Autoregressive Infilling Algorithm

Algorithm 2 describes the baseline autoregressive infilling algorithm evaluated in Table 3. Without the ability to anticipate future controls, this algorithm proceeds by sampling from the model until the time of the sampled event time exceeds the time of the next control. At this point, we insert this control (and any other controls prior to the sampled event) into the sequence prior to the sampled event. We then proceed to continue sampling from the model.

---

**Algorithm 2:** Autoregressive Sampling (Baseline)

---

**Parameters:** Autoregressive model $p$ with context length $M$
**Input**       : Time-localized controls $\mathbf{u}_{1:K}$ (monotone increasing in time)
**Output**      : A generated sequence $\mathbf{a}_{1:N+K}$

$\mathbf{a}_0 \leftarrow$ SEP ;                                              // A special sequence separator event
$i \leftarrow 1$ ;                               // Index $i$ tracks position in the generated sequence
$k \leftarrow 0$ ;                               // Index $k$ tracks position in the control sequence
**do**
  Sample $\mathbf{e} \sim p(\cdot | \mathbf{a}_{i-M:i-1})$ ;                              // Sample an event from the model
  $\mathbf{t} \leftarrow \text{Time}(\mathbf{e})$ ;                                    // Get the time $\mathbf{t}$ of the event $\mathbf{e}$
  **while** $\text{Time}(\mathbf{u}_k) \leq \mathbf{t}$ ;                          // While there are controls before time $\mathbf{t}$
   **do**
     $\mathbf{a}_i \leftarrow \mathbf{u}_k$ ;                                      // Anticipate control $\mathbf{u}_k$ at index $i$
     $i \leftarrow i + 1$ ;                                            // Advance to index $i + 1$
     $k \leftarrow k + 1$ ;                                            // Consume control $\mathbf{u}_k$
   **end**
  $\mathbf{a}_i \leftarrow \mathbf{e}$ ;                                  // Append the newly sampled event
  $i \leftarrow i + 1$ ;                                            // Advance to index $i + 1$
**while** $\mathbf{a}_i \neq$ SEP;
**return** $\mathbf{a}_{1:i-1}$ ;                                        // Index $i - 1$ is $N + K$

---

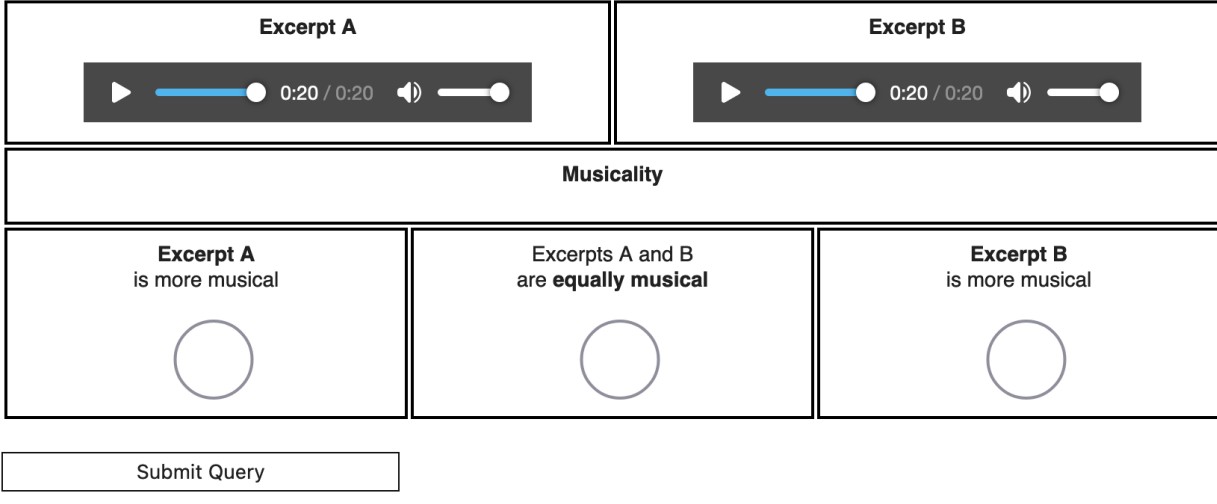

Figure 4: The interface used by evaluators to assess the relative musicality of paired music clips.

This algorithm is naive because the model cannot condition on upcoming controls until after the time at which they occur. In the most optimistic scenario, because the model is a good predictor of events, we hope that it will assign high probability to the upcoming control events and therefore that the model will generate events that are consistent with (reasonable, unsurprising) controls. But ultimately there is entropy in the music process, and the model will necessarily sometimes generate events that are inconsistent with upcoming controls. To make matters worse, these inconsistencies are then written into the history and the model makes subsequent predictions conditioned on these mistakes; this has a tendency to compound the errors over time, creating highly dissonant music.

A more subtle instance of this failure of compounding errors is the algorithm's tendency to double the control events. It is relatively common for the model will exactly predict an upcoming control event. In this case, two copies of the event are written into the history. When subsequent controls are inserted into the history, the model exhibits a tendency to perpetuate this doubling, generating new events that copy every control that we insert into the history, like a vexing child copying a sibling's every word.

## J Details of Human Evaluation

Workers chosen to evaluate generated music were selected from a pool of crowd workers on the Amazon Mechanical Turk platform, according to the qualification procedure described below. Evaluators were provided with the interface shown in Figure 4 and instructed to judge the relative musicality of two 20-second music clips. We presented the music as audio, synthesized using Apple's DLSMusicDevice sound system. Each pair of clips begins with the same prompt: three bars for the prompted completion task and five seconds for the accompaniment task. Evaluators were instructed to judge which clip is more musical. Based on feedback from a pilot study, we clarified in the detailed instructions that we are interested in musicality in a *conventional* sense. We allowed evaluators to indicate that the two clips are equally musical, avoiding a forced choice between the two clips.

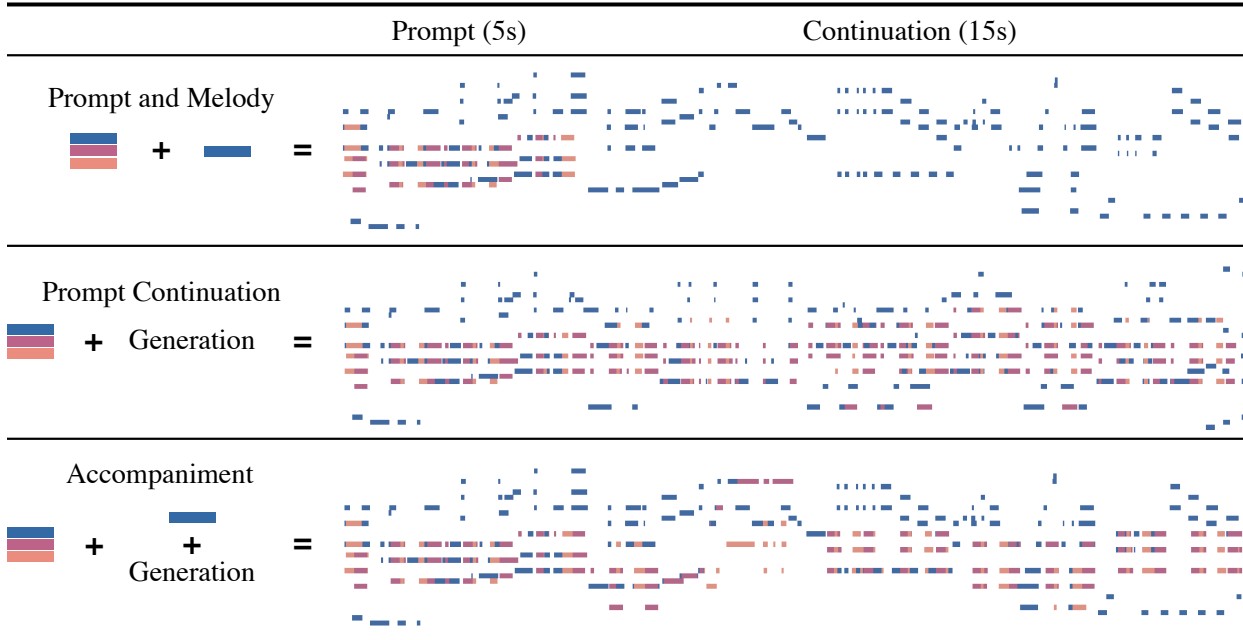

Figure 5: Visualizations of 20-second music clips. Each rectangle indicates a musical event with an onset time, duration (width), and pitch (height). Colors indicate distinct instrumental parts. For the accompaniment task, events in the blue instrumental part are used as control events. Top: a five-second prompt followed by the original continuation of only the melodic instrumental line (highest; blue). Middle: the five-second prompt followed by a generated autoregressive continuation, ignoring the original melodic line. Bottom: the prompt followed by a generated anticipatory accompaniment of the original melodic instrumental line.

All prompts and melodies used for evaluation are sampled randomly from the test set. A notable confounder in this evaluation is that the Lakh MIDI contains many popular, recognizable songs. Study participants remarked that they recognized the origin of certain prompts. In these cases, we instructed the participants to "Please try to rate the clips based on their musicality rather than recognition." However, it may be difficult to set aside knowledge of the canonical completion in these cases.

Details of qualification, prompt continuation, and accompaniment evaluations are described below. For an illustration of the prompt continuation and accompaniment tasks, see Figure 5. For complete pairwise evaluations of models in the prompt continuation task, see Table 10. For complete pairwise evaluations of algorithms in the accompaniment tasks, see Table 11. For each task, we collected evaluations from three unique workers; we defer an analysis of worker agreement to future work.

**Qualification** Workers who correctly identified music composed by humans as more musical than random retrieval in at least 4/5 comparisons were deemed qualified to participate in subsequent model evaluations: 15 out of 20 workers who participated in qualification advanced to the main prompt continuation and accompaniment evaluations. Workers performed a total of 100 comparisons.

**Prompt Continuation** We compare continuations of four models: (i) the Small interarrival-time model (Row 1, Table 1) (ii) the Small anticipatory model (Row 3, Table 1) (iii) the large anticipatory model (Row 5, Table 1) and (iv) the FIGARO Music Transformer. We also compare to (v) human compositions (skyline). For each of 50 prompts, we create $50 \times \binom{5}{2} = 500$ pairwise comparisons between continuations (as well as the baseline and skyline) we asked three human evaluators to indicate which clip is more musical, or that the clips are equally musical. Workers performed a total of 1500 comparisons.

**Accompaniment** We compare accompaniments using (i) anticipatory autoregressive sampling (Algorithm 1) versus (ii) baseline autoregressive sampling. We also compare to (iii) baseline completions randomly sampled from the test set and (iv) skyline human compositions. When sampling without anticipation, we

insert events from the melody into the conditional history of the model once generation passes them; we describe this modified sampling procedure formally in Appendix I. For each of 20 three-bar prompts and single-part continuations, we generated accompaniments from each model. For each of the $20 \times \binom{4}{2} = 120$ pairwise comparisons between continuations (as well as the baseline and skyline) we asked three human evaluators to indicate which clip is more musical, or that the clips are equally musical. Workers performed a total of 360 comparisons.

Table 10: Human evaluation of paired completions of 3-bar musical prompts generated by different algorithms, and human compositions. P-values are reported using a Wilcoxon signed rank test. Row numbers reference Table 1.

| Model A | Model B | Wins (A) | Ties | Wins (B) | p-value |
|---|---|---|---|---|---|
| Human Composition | Medium (Row 8) | 77 | 29 | 44 | 0.0027 |
| | Music Transformer | 119 | 18 | 13 | $2.806 \times 10^{-20}$ |
| | Small (Row 3) | 84 | 23 | 43 | 0.0002 |
| | Small (Row 1) | 103 | 16 | 31 | $4.976 \times 10^{-10}$ |
| Medium (Row 8) | Music Transformer | 95 | 24 | 31 | $1.187 \times 10^{-08}$ |
| | Small (Row 3) | 65 | 27 | 58 | 0.528 |
| | Small (Row 1) | 96 | 17 | 37 | $3.122 \times 10^{-7}$ |
| Music Transformer | Small (Row 3) | 36 | 16 | 98 | $8.509 \times 10^{-08}$ |
| | Small (Row 1) | 46 | 18 | 86 | 0.0005 |
| Small (Row 3) | Small (Row 1) | 82 | 17 | 51 | 0.0071 |

Table 11: Human evaluation of paired 15-second accompaniments generated by different models, and human-composed accompaniments. P-values are reported using a Wilcoxon signed rank test.

| Algorithm A | Algorithm B | Wins (A) | Ties | Wins (B) | p-value |
|---|---|---|---|---|---|
| Human Composition | Anticipatory | 11 | 31 | 18 | 0.194 |
| | Autoregressive | 45 | 10 | 5 | $1.542 \times 10^{-08}$ |
| | Retrieval | 52 | 6 | 2 | $1.017 \times 10^{-11}$ |
| Anticipatory | Autoregressive | 47 | 6 | 7 | $5.230 \times 10^{-8}$ |
| | Retrieval | 45 | 11 | 4 | $4.709 \times 10^{-9}$ |
| Autoregressive | Retrieval | 33 | 12 | 15 | 0.009 |

## K   Training Optimization Logs

Figure 6 and Figure 7 plot estimates of the train set and test-set losses over the course of optimization for the arrival-time models considered in Section 4. Losses are computed every 10,000 steps from logged model checkpoints; the intermediate checkpoints for all 8 of these models (as well as the small interarrival-time model) are available on request.

Training loss is significantly lower than test for all models, evidence of some amount of overfitting during many epochs of optimization on the Lakh MIDI dataset. Nevertheless, we observe better test set performance for larger models, trained longer, indicating that we have not completely saturated performance on the Lakh MIDI dataset using this scale of models and computational resources. That said, the relative test set loss improvements vs train set improvements when we increase the model size from Small to Medium are much larger than the relative gains of increasing the model size from Medium to Large: compare train vs. test loss of the Small (Row 3) Medium (Row 6) and Large (Row 9) models at 100k steps. This might suggest that we are approaching the point of diminishing returns for scaling the compute (steps) and size of models trained on Lakh MIDI, and that more data would be needed to effectively train substantially larger models.

For the Small models, we observe that autoregressive training results in slightly better test-set performance at 100,000 optimization steps. But at 800,000 steps the situation reverses, and the anticipatory model performs slightly better. We suspect that this is evidence of anticipatory training having a regularizing effect on the optimization. The Small, 800k step anticipatory model also exhibits some training instability in these plots. We observed similar instabilities during training the other anticipatory and autoregressive models, but they do not appear at the 10,000-step granularity pictured here. We undertook no measures to adjust for these instabilities, simply letting the optimizations run to completion without intervention.

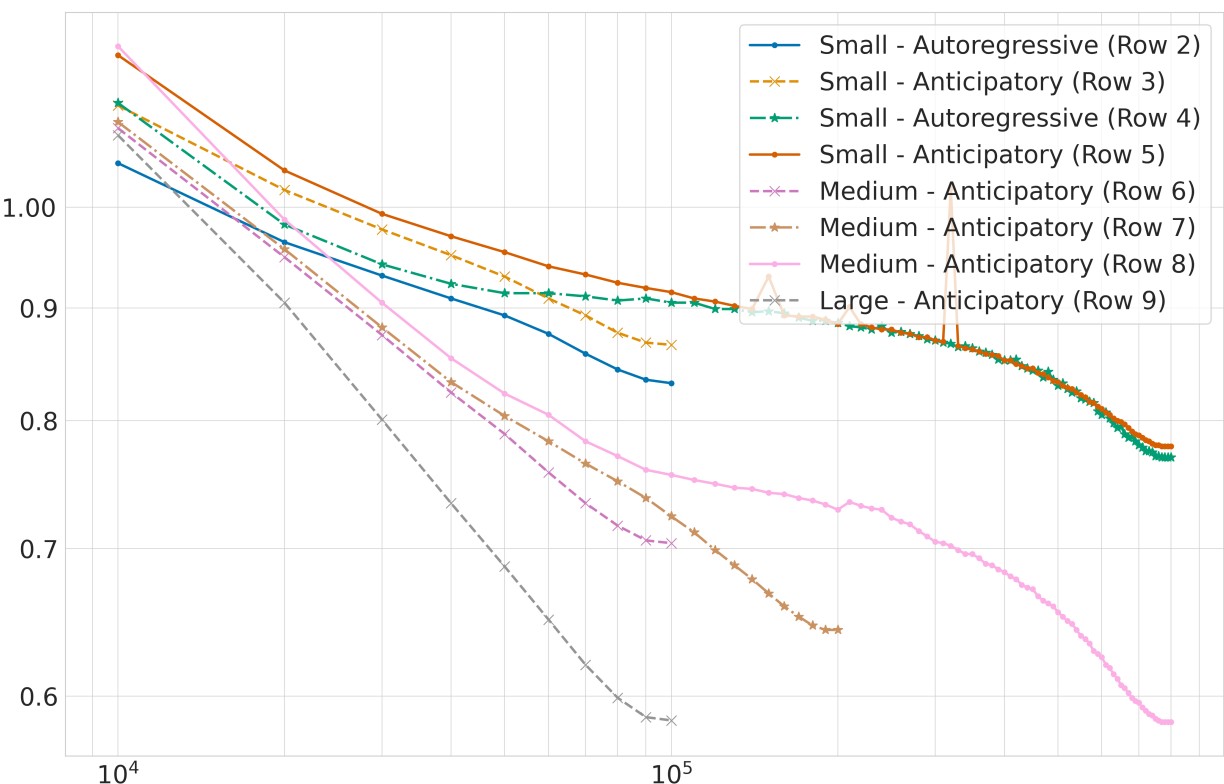

Figure 6: Mean per-token train set log-loss $L_{\text{token}}$ (in nats) of models, estimated every 10,000 steps over the course of training. The estimates are computed using a 1/100 subset of the train set. The per-token loss $L_{\text{token}}$ is related the event loss reported in Table 1 by the relationship $L_{\text{token}} = \log(L_{\mathbf{e}})/3$.

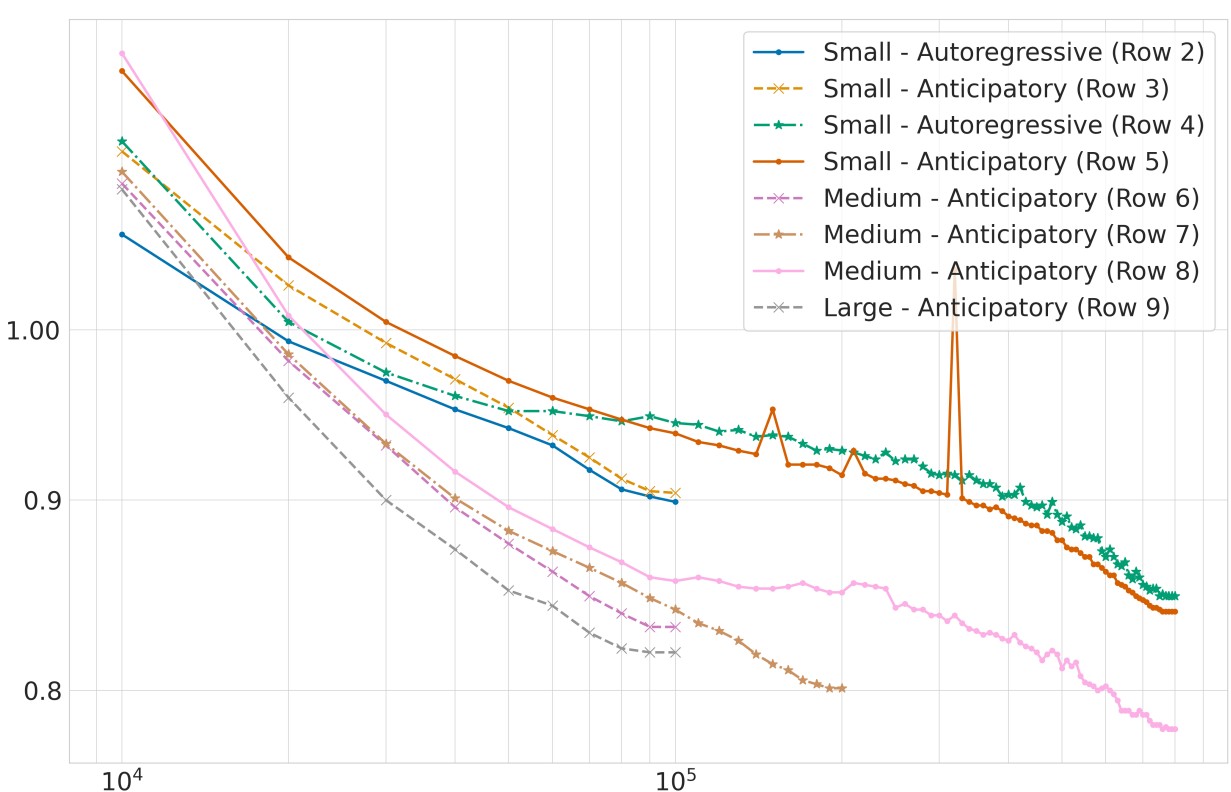

Figure 7: Mean per-token test set log-loss $L_{\text{token}}$ (in nats) of models, estimated every 10,000 steps over the course of training. The estimates are computed using a 1/10 subset of the test set. The per-token loss $L_{\text{token}}$ is related the event loss reported in Table 1 by the relationship $L_{\text{token}} = \log(L_{\mathbf{e}})/3$.

## L  Model Card

Table 12: Model Card (Mitchell et al., 2019) - Anticipatory Music Transformer.

| **Model Details** | |
| --- | --- |
| Organization Developing the Model | Stanford Center for Research on Foundation Models |
| Model Date | June 2023 |
| Model Type | Autoregressive Causal Transformer |
| Additional Modeling Details | See Section 3 |
| License | Apache License, Version 2.0 |
| Correspondence | jthickstun@cs.stanford.edu |
| **Intended Use** | |
| Primary Intended Uses | Collaborative co-composition between a human composer and an Anticipatory Music Transformer. The role of the anticipatory model in this collaboration could include, e.g., infilling tedious/low-entropy details (productivity enhancement) and suggesting possible continuations (creative ideation). |
| Primary Intended Users | Artists, musicians, and composers. |
| Out-of-Scope Uses | **Long-Context Generation.** These models cannot generate full-length song structures without human control. The models have a context length of 1024 tokens (331 events). At 68 tokens/second (the average for Lakh MIDI; see Appendix E) this corresponds to approximately 15 seconds of context. Models conditioned on more than 331 events will only use the most recent 331 events (including anticipated events) to predict the next event.
**Music Metadata.** These models do not explicitly model or generate metadata, including: metrical structure, key signature, tempo, note-value (eighth-note, quarter-note etc.).
**Extended Music Vocabulary.** These models generate sequences with a narrow vocabulary of notes, instruments, and timings. They do not model or generate other aspects of music, including: dynamics, articulations, or lyrics. |
| **Factors** | |
| Western Bias | These models are trained on the Lakh MIDI dataset, a collection of predominantly Western music. See Section A.3 for further discussion. |
| **Metrics** | |
| Automatic Metrics | Next-event perplexity (defined in Table 1) and bits per seconds (defined in Appendix F). |
| Human Evaluation | Pairwise human preferences between generated music and reference compositions. |
| Decision Thresholds | For human evaluation, we generated samples from anticipatory models using nucleus sampling with $p = 0.95$. See Section 4 for further discussion. |

| Approaches to uncertainty and variability | We report p-values for pairwise comparisons between music generated by different models and ground truth music using the Wilcoxon signed-rank test. Due to computational constraints, we do not account for variability in the model training process, such as dataset splits or the random seed for optimization. |
|---|---|

**Datasets**

| Training Data | The `0`–`d` splits of the Lakh MIDI dataset, augmented using anticipation (see Section 3) with the prior distribution over controls described in Appendix D. |
|---|---|
| Validation Data | The `e` split of the Lakh MIDI dataset. |
| Test data | The `f` split of the Lakh MIDI dataset. |
| Out-of-Distribution Data | We do not evaluate out-of-distribution performance. |
| Preprocessing | Preprocessing and filtering of the Lakh MIDI dataset is described in Appendix E. |
| Motivation | We chose to work with the Lakh MIDI dataset because it is the largest collection of symbolic music data currently in use by the machine learning community. |

**Quantitative Analyses**

| Aggregated Analysis | Our analysis of aggregate results based on automatic metrics and human evaluation are presented in Section 4.1 and Section 4.2 respectively. Key findings include:

• Anticipatory training does not interfere with autoregressive model performance, as measured by perplexities of comparable anticipatory and autoregressive models.

• Accompaniments generated by an Anticipatory Music Transformer have similar musicality to ground truth accompaniments according to human evaluators. |
|---|---|
| Disaggregated Analysis | We do not perform a disaggregated analysis of the Anticipatory Music Transformer. One obstruction to conducting such an analysis is a lack of metadata associated with the Lakh MIDI dataset. |

**Ethical Considerations**

| Labor Displacement | We are broadly concerned by the transient disruptions of labor markets caused by the introduction of new productivity-enhancing and automative technologies. See Section A.1 for a discussion of the possible disruptive effects of generative music models on the creative economy. |
|---|---|
| Copyright | The Lakh MIDI dataset contains large quantities of copyrighted music. The copyright status of models trained on this data—and music sampled from these models—is an open legal question. See Section A.2 for further discussion. |

