# OpenReview forum: "Anticipatory Music Transformer"
_TMLR — Accepted by TMLR_

### Review · Reviewer_ynfy · 2023-11-27

**Summary Of Contributions:**

The authors present anticipation, a novel encoding for temporal point processes to be used with autoregressive sequence modeling. They apply it to the task of infilling in symbolic music (demonstrating continuation and accompaniment generation capabilities).

**Audience:**

Yes

**Broader Impact Concerns:**

The authors comprehensively acknowledge the ethical concerns of generative music modeling in the appendix, to a much greater extent than I have seen in related literature.

**Claims And Evidence:**

Yes

**Requested Changes:**

1. Small grammatical error on page 3, third paragraph. "This generalizes the span-infilling task--which *asks* us..."
2. The "Related Work on Musical Accompaniment" section is under 4.2, and there is another entire section (section 5) for related work on controllable generative modeling. It might make more sense to group all related work discussions under one section and differentiate within with subsections.

**Strengths And Weaknesses:**

### Strengths
* The authors' use of examples throughout section 3 were very helpful in illustrating how the interleaving of events occurs.
* The authors provided detailed informaton about their methodology with regards to human evaluation.
* The conclusion does a good job connecting the ideas in the paper to the broader idea of using locality to improve performance of machine learning models. I also appreciate the authors discussion of the importance of controllability for facilitating the adoption of generative music models by artists and creatives.

### Weaknesses
* I wonder why the authors did not compare to Music Transformer for the accompaniment task since that model (at least the original) can also be used for accompaniment generation.

---

> ### Author Response · Authors · 2024-01-28
> **Followup Response to Reviewer ynfy**
>
> Thank you again for your feedback on our paper. We would like to know whether we have adequately responded to your feedback. We would also like to share an additional observation (below) in response to your question about using Seq2Seq models for tasks like accompaniment.
>
> In addition to the technical motivations for anticipation discussed in our earlier comment, anticipation can also be motivated from the perspective of interaction design. A Seq2Seq model requires that all the control tokens be specified in advance (because they must be placed at the beginning of a sequence). For example, in the accompaniment task, this means that a user must specify a complete melody before the model can suggest an accompaniment. This supports a simple two-stage collaborative process, whereby the user writes a melody followed by the model generating an accompaniment.
>
> In contrast, because an anticipatory model sees a limited distance into the future--5 seconds for the models trained in our paper--an anticipatory model can begin to suggest an accompaniment as soon as the user has composed at least 5 seconds of the melodic line. Anticipatory model could therefore support more interactive collaborative processes than Seq2Seq, whereby the user begins to compose a melody, the model suggests an accompaniment for the first few bars, which in turn inspires the user's expansion of the melodic idea in future bars, etc. We are excited to explore these human-computer interaction questions, facilitated by anticipation, in future work!

---

> > ### Comment · Reviewer_ynfy · 2024-02-12
> > **Response to authors**
> >
> > Thank you for addressing my comments. Your explanation of the distinction between the traditional Seq2Seq model and an anticipatory model makes it clear why it is not pertinent to compare the two types of models. The tasks are different enough that the comparison would not be particularly meaningful.

---

### Review · Reviewer_Wdtb · 2023-12-01

**Summary Of Contributions:**

This paper proposes a new method, denoted as anticipation, which allows the construction of a controllable generative model. The proposed approach is based on modeling several parallel sequences conditioned asynchronously. The authors presented results on symbolic music generation (MIDI) and showed that the proposed method reached superior results than the standard auto-regressive modeling. Additionally, the authors present results on accompaniment MIDI generation and show that the proposed method is superior to both auto-regressive and retrieval-based baseline.

**Audience:**

Yes

**Claims And Evidence:**

Yes

**Requested Changes:**

**Questions to the authors:**

1. Can the authors present results / demonstrate the applicability of the proposed approach on other tasks than MIDI generation? As stated above, it seems the solution and input representations are heavily tailored to MIDI, which might not be of high interest to the ML community at large. It is not clear whether this approach can be applied to other domains/tasks as well.

2. In the FIGARO paper [1], the authors reported objective metrics alongside human study. Can the authors additionally provide objective results? It would make it easier to compare with prior work and will make the paper stronger.

**Strengths And Weaknesses:**

**Strengths:**
1. The proposed modeling approach is interesting and novel.
2. The paper is clearly written and the notations are well defined.
3. Results suggest the proposed method is superior to the evaluated baselines.

**Weakness:**
1. It seems the authors designed a solution that is specifically tailored to the task of MIDI generation. It is not clear whether this will be of interest to the ML community at large.
2. The comparison to the baseline methods considers human evaluation only.

---

> ### Author Response · Authors · 2024-01-28
> **Followup Response to Reviewer Wdtb**
>
> Thank you again for your feedback on our paper. Do the quantitative metrics we have provided satisfy your request for objective results? And have we adequately addressed your concern about broader interest in anticipation by the ML community? Generalizing the speech example given in our previous response (which we believe to be quite promising!) we can imagine many applications of anticipation for controllable generation/simulation of time-aligned, asynchronous modalities. To provide some additional examples: video + scripts, audio + lyrics, social media activity + world events.

---

> > ### Comment · Reviewer_Wdtb · 2024-02-01
> > **Response to the authors**
> >
> > I would like to thank the authors for addressing my comments. I believe the objective results make the contribution of this paper stronger.
> >
> > Regarding the additional tasks, I'm not sure I understand what the authors mean by "speech example given in our previous response"? I might have missed that but I did not see any speech examples.
> >
> > I appreciate the author's response about the additional tasks/applications that can leverage the proposed method, I think the authors should include it in the paper in the introduction/discussion section. Having said that, I still encourage the authors to include additional results for other domains/setups (maybe one of the setups proposed by the authors) that can demonstrate the efficiency of the proposed method beyond MIDI representation.

---

> > > ### Author Response · Authors · 2024-02-02
> > > **Response**
> > >
> > > > Regarding the additional tasks, I'm not sure I understand what the authors mean by "speech example given in our previous response"? I might have missed that but I did not see any speech examples.
> > >
> > > In our initial response, we provided several examples of non-music temporal point processes, along with a specific example that we believe would be a fruitful application of anticipation. We have reproduced that discussion below.
> > >
> > > Temporal point processes are of broad interest to the machine learning community, with applications to modeling social media [7], clinical data [8], e-commerce [9], and neuroscience [10], among other domains. As a concrete example: time-aligned text is a temporal point process. We can imagine applying anticipation to generate audio transcripts of speech, where the model anticipates the next few seconds of audio when generating the transcript.
> > >
> > > > I appreciate the author's response about the additional tasks/applications that can leverage the proposed method, I think the authors should include it in the paper in the introduction/discussion section. Having said that, I still encourage the authors to include additional results for other domains/setups (maybe one of the setups proposed by the authors) that can demonstrate the efficiency of the proposed method beyond MIDI representation.
> > >
> > > We will add a discussion of the broader potential applicability of anticipation to the paper.
> > >
> > > While we appreciate the reviewers interest in broader applications of anticipation--and we believe that many applications could be quite fruitful--pursuing these applications are clearly beyond the scope of the present work, which focuses on music. We emphasize that the music domain is of long-established interest to the ML community [1,2,3].
> > >
> > > In addition to the methodological contribution of anticipation, our paper makes multiple contributions to the music domain including:
> > >
> > >   * the construction & release of pre-trained models that required over 1100 hours of time on a TPU v3-32 machine to train (Appendix G).
> > >   * an arrival-time tokenization of music (Section 2; Appendix C).
> > >   * experimental design & human evaluation of these models (Section 4; Appendix J)
> > >   * a discussion of ethical considerations for generative modeling in the music domain (Appendix A)
> > >
> > > Applications of anticipation in another domain would be better suited to a separate paper, which could more fully explore the analogs of these contributions.
> > >
> > > [1] Allan and Williams. Harmonising chorales by probabilistic inference. Neurips, 2004.
> > >
> > > [2] Boulanger-Lewandowski, Bengio, and Vincent. Modeling temporal dependencies in high-dimensional sequences: application to polyphonic music generation and transcription. ICML, 2012.
> > >
> > > [3] Huang, et al. Music Transformer: Generating Music with Long-Term Structure. ICLR, 2018.

---

### Review · Reviewer_9MMW · 2024-01-15

**Summary Of Contributions:**

The paper proposes a novel model---anticipatory music Transformer---for the task of infilling control.
The model relies on a novel tokenization and shifting technique, such that the generation of an accompaniment note will depend on not only past notes but also nearby future melody notes.

The model achieves compelling results on Lakh MIDI dataset, under automatic and human evaluation.

All code and pretrained model weights will be released.

**Audience:**

Yes

**Broader Impact Concerns:**

No ethical concerns.

**Claims And Evidence:**

Yes

**Requested Changes:**

Please correct me if I am wrong.

First, the term "point process" should describe the model, not the data.
Precisely, a point process is a generative model of sequences of time-stamped events. So phrases like the following are wrong:
- "point process consists of time-ordered events" (Def-2.1)
- "generative models of a temporal point process" (Sec-1)

Second, it doesn't seem to be necessary to emphasize "point process" because the key technique in this paper is to tokenize the sequence such that one can use LM objective, and there is no notion of "intensity" in the proposed method, which is a key element in point processes.

Third, the part about "stoping time" seems unnecessarily complicated. Some stochastic calculus terms (e.g., "filtration") are introduced but do not really help readers understand the essence.
As I understand, "stoping time" is important because the predicted "events" may go out of the boundary, passing the "controls" that actually should happen after them. So the method must be aware of the "stoping time" during generation, is it right?
I think this point can be made clear without involving any more sophisticated concepts (e.g., "filtration").

Generally, I think the paper will be much shorter and clearer if the complication above can be resolved.

**Strengths And Weaknesses:**

Strengths

The tokenization and shifting design is a clever idea and its experiment results are compelling.

The code and pretrained model weights will be a useful contribution as well once they are released.

Weaknesses

There are 3 key weaknesses: missing references, missing comparison, and unclear presentation.

Missing references.

Hongyuan Mei, Guanghui Qin, Jason Eisner. 2019. Imputing Missing Events in Continuous-Time Event Streams.

This paper proposes a particle smoothing method that predicts unobserved events based on past observed and unobserved events as well as future observed events.
This smoothing method is similar to the proposed method except that it doesn't have the "nearby" inductive bias; the "nearby" bias seems very reasonable for music specifically.

Missing comparison.

The above method is a natural baseline to compare with.

Unclear presentation.

The paper is very hard to follow. The term "point process" is a little abused and the presentation relies upon unnecessarily complicated formulation (e.g., filtration).

I think the presentation could and should be significantly simplified before it is published.

Please see Requested Changes for my detailed notes.

---

> ### Author Response · Authors · 2024-01-21
> **Response to Reviewer 9MMW**
>
> Thank you for your feedback. We appreciate your positive comments regarding the compelling results of our models, and the value of our code and pre-trained models for the community. We also appreciate that you find our anticipatory modeling technique to be clever.
>
> We first address your comments related to neural Hawkes processes, then address your questions regarding the presentation of results (in another reply, due to space limitations).
>
> > there is no notion of "intensity" in the proposed method, which is a key element in point processes.
>
> You are familiar with literature on temporal point processes that parameterize and model the conditional intensity of a process. This parameterization is common in the machine learning literature (e.g., the neural Hawkes process). As we remark in the beginning of our related work (Section 5) we take a different approach: non-parametric density estimation via maximum likelihood estimation. While we do not model a conditional intensity function, we nevertheless model a temporal point process, and the structure of this process is critical to our work. Shchur et al. [1] discusses some of the tradeoffs and considerations for modeling using intensity functions vs. densities.
>
> > Hongyuan Mei, Guanghui Qin, Jason Eisner. 2019. Imputing Missing Events in Continuous-Time Event Streams.
>
> Mei et al. consider temporal point processes with simple mark structure (e.g., taxi pickups and elevator systems). In each case, the mark of the process is a simple categorical item taking just one of K=10 values. By contrast, marks in the music setting–musical notes–have rich structure: pitch, instrument, duration. It is unclear to us how we would adapt Mei et al.’s methodology to music. Mei et al. model a neural Hawkes process, parameterizing each distinct mark category by a conditional intensity function. For music, treating each mark as a category would result in over 16 million categories, each of which must be modeled by a distinct intensity function. Modifying the work of Mei et al. to apply their methods in our setting is beyond the scope of this paper.
>
> We also remark that our method is quite conceptually quite different from Mei et al. They approximate samples of missing data using sequential Monte Carlo (SMC), given a joint model trained on complete sequences. They leverage the structure of the Hawkes process to construct a proposal distribution for SMC approximation. In contrast, we assume no such structure and directly model conditional distributions over missing events using anticipation, obtaining exact samples of missing data from the trained model.
>
> [1] Intensity-free Learning of Temporal Point Processes. Oleksandr Shchur, Marin Bilos, Stephan Gunnemann. ICLR, 2020.

---

> > ### Comment · Reviewer_9MMW · 2024-02-11
> > **rebuttal misses my key points**
> >
> > re. intensity-free or not
> >
> > I can write long paragraphs about why "intensity" is a key notion for "point processes", but I rather be super concise here about my key point, which the rebuttal seems to miss: calling your model as a "point process" is not necessary or helpful for readers to understand the paper at all, then why not do so? Characterizing the distribution of x without specifying intensity is just like constructing a distribution over (all kinds of) sequences, right? Then why is not the model also called a kind of "language model", which learns to predict the next token given the context, without specifying intensity at all? Calling it as "Anticipatory Transformer" is just technically appropriate, that is my point.
> >
> > re. compare with Mei et al.
> >
> > First, they didn't "leverage the structure of the Hawkes process" at all; instead, the method is generic and works for neural models in their experiments.
> >
> > Second, their key point is to estimate the distribution over missing data using a bidirectional model, which is very different from your proposed method, but the task is the same---imputing missing events. Is it just the right reason for you to compare with it? Other things you pointed out about their method is superficial, and thus doesn't justify a no-comparison.

---

> > > ### Author Response · Authors · 2024-02-19
> > > **regarding sequential monte carlo methods for infilling**
> > >
> > > > re. compare with Mei et al. [...] their key point is to estimate the distribution over missing data using a bidirectional model, which is very different from your proposed method, but the task is the same---imputing missing events.
> > >
> > > While we cannot run the exact sequential Monte Carlo (SMC) sampler proposed by Mei et al. for the aforementioned reasons, we could certainly imagine comparing our methods to an SMC sampler inspired by Mei, using a right-to-left model over controls to construct a proposal distribution over the next event.
> > >
> > > One obstruction to this approach is that the sequences modeled in our paper are too long to fit into a model's context. The longest sequence length considered in Mei's paper is 370 tokens (Table 1 of Mei et al.) whereas the average length of a sequence considered in our paper is 12,071 tokens (Figure 2 of our paper). Therefore, we would still need to apply some truncation heuristic for selecting a context of past + future controls for proposing events.
> > >
> > > > [Mei's] smoothing method is similar to the proposed method except that it doesn't have the "nearby" inductive bias; the "nearby" bias seems very reasonable for music specifically.
> > >
> > > Once we choose a heuristic for truncation, an SMC sampler for our data would also exhibit a nearby bias, just like anticipation. Indeed, it is not obvious what heuristic to use for truncation other than anticipation. Both Seq2Seq modeling (discussed in the paper) and bidirectional SMC modeling (like Mei et al.) face the same problem with respect to long sequences. Our proposal to manage long context using anticipation is orthogonal to whether we build a model that supports exact inference (e.g., Seq2Seq) or approximate SMC inference (e.g., Mei et al.).
> > >
> > > We agree that SMC approaches to infilling merit further discussion, and we will update our related work section to discuss Mei et al.'s work and other SMC methods in the next version of our paper.

---

> > > > ### Comment · Reviewer_9MMW · 2024-02-19
> > > > **still misunderstanding**
> > > >
> > > > > While we cannot run the exact sequential Monte Carlo (SMC) sampler proposed by Mei et al. for the aforementioned reasons
> > > >
> > > > Aforementioned reasons are not valid, are they? E.g., that SMC algorithm is not exact, but approximate.
> > > > (btw., how can an SMC algorithm be exact in any case?)
> > > >
> > > > > we could certainly imagine comparing our methods to an SMC sampler inspired by Mei, using a right-to-left model over controls to construct a proposal distribution over the next event.
> > > >
> > > > That sounds just the Mei et al's method, what is the difference?
> > > >
> > > > > One obstruction to this approach is that the sequences modeled in our paper are too long to fit into a model's context.
> > > >
> > > > This seems a valid point. But questions are (please answer concisely):
> > > > - How did your method handle long sequences?
> > > > - Did you do anything specific (sorry if I miss it) to make your method scalable to long sequences but it is not done by Mei et al. 2019?
> > > >
> > > > > We agree that SMC approaches to infilling merit further discussion
> > > >
> > > > That is my point!
> > > > As I said, I like your method and results! But there are other kinds of methods that also naturally fit in this problem and they should be appropriately discussed.

---

> > > > > ### Author Response · Authors · 2024-02-19
> > > > > **re: obstructions to running Mei's method**
> > > > >
> > > > > > Aforementioned reasons are not valid, are they? E.g., that SMC algorithm is not exact, but approximate. (btw., how can an SMC algorithm be exact in any case?)
> > > > >
> > > > > Apologies: "exact" was a poor choice of words in this context. I should have said "we cannot run the Monte Carlo (SMC) sampler proposed by Mei et al. _without significant modifications_." I will attempt to clarify the aforementioned obstruction to running Mei's method out-of-the-box below. That said, on reflection, we believe that our point about sequence length and anticipation is a more fundamental obstruction to running methods like Mei's, as this issue arises for any bidirectional approach to modeling (including other forms of SMC, as well as exact inference methods like Seq2Seq).
> > > > >
> > > > > > "we could certainly imagine comparing our methods to an SMC sampler inspired by Mei, using a right-to-left model over controls to construct a proposal distribution over the next event."
> > > > > > That sounds just the Mei et al's method, what is the difference?
> > > > >
> > > > > The difference is in whether we model conditional intensities, or model sequences. Mei parameterizes a conditional intensity function over each category of mark, whereas our model parameterizes a predictor of the next token in a sequence of serialized events.
> > > > >
> > > > > Directly parameterizing the conditional intensity of each mark is impractical in our setting: I am referring to Equations (2) and (5) of Mei's paper. The music data has a large vocabulary of K = 16.5 = 1000 * 128 * 129 million marks (the cross product of 1000 possible durations, 128 possible pitches, and 129 possible instrument classes) and the v_k matrices appearing in these equations simply cannot be materialized (the sums in Line 25 of Algorithm 1 are also daunting).
> > > > >
> > > > > In contrast, our sequence models handle the large vocabulary of marks by factoring the distribution over marks into "sub-word tokens" for duration, pitch and instrument; this keeps the vocabulary size manageable. We could imagine constructing a hybrid model, that autoregressively factors the distribution over conditional intensities, but this alteration is open-ended, and diverges significantly from the methods proposed by Mei et al.
> > > > >
> > > > > Another approach to SMC, that retains the spirit of Mei's algorithm but in the sequence modeling setting, would use the model we have already trained as the forward, left-to-right model. We could then train a backward, right-to-left model that predicts sequences of control tokens in reverse order to construct a proposal distribution. This approach might bear more similarities to [1] than to Mei et al.
> > > > >
> > > > > As mentioned previously, we will add a discussion of these and other SMC approaches to our related work.
> > > > >
> > > > > [1] Neural Particle Smoothing for Sampling from Conditional Sequence Models
> > > > > Chu-Cheng Lin and Jason Eisner

---

> > > > > > ### Comment · Reviewer_9MMW · 2024-02-20
> > > > > > **on same page for comparing with particle smoothing**
> > > > > >
> > > > > > > ... I will attempt to clarify the aforementioned obstruction to running Mei's method out-of-the-box below. That said, on reflection, we believe that our point about sequence length and anticipation is a more fundamental obstruction to running methods like Mei's, ...
> > > > > >
> > > > > > Good point. I never mean that you should run their code off-the-shelf. I mean comparing that *kind* of methods to understand the pros and cons of each paradigm.
> > > > > >
> > > > > > > Directly parameterizing the conditional intensity of each mark is impractical in our setting... In contrast, our sequence models handle the large vocabulary of marks by factoring the distribution over marks into "sub-word tokens" for duration, pitch and instrument...
> > > > > >
> > > > > > This makes sense.
> > > > > > The key reason that I wanted you to compare with Mei et al. 2019 is to compare your method with another general paradigm that treats event imputation in a straightforward way, like SMC.
> > > > > > It is okay that you keep the essential parts of that paradigm but skip the superficial details, like how to set up vocab.
> > > > > >
> > > > > > > another approach to SMC, that retains the spirit of Mei's algorithm but in the sequence modeling setting, ... This approach might bear more similarities to [1] than to Mei et al.
> > > > > >
> > > > > > I think these 2 papers are the same except that they work on different kinds of data (and thus their "probabilities" are different).
> > > > > > But this is not crucial.

---

> ### Author Response · Authors · 2024-01-21
> **Response to Reviewer 9MMW (continued)**
>
> > First, the term "point process" should describe the model, not the data. Precisely, a point process is a generative model of sequences of time-stamped events.
>
> A temporal point process is a probability distribution with a particular structure. Both the data-generating distribution (from which data is sampled) and a generative model fit to this data have the structure of a temporal point process. The events in Section 3 are random variables–-not data–that denote this structure. By “generative models of a temporal point process,” we mean models trained to approximate the distribution of training data sampled from a temporal point process.
>
> > Second, it doesn't seem to be necessary to emphasize "point process" because the key technique in this paper is to tokenize the sequence such that one can use LM objective.
>
> The key technique of our paper is not to tokenize a sequence for use of the LM objective, but rather it is anticipation: a way to interleave two asynchronous sequences such that we can generate one sequence, conditioned on the other. The point process structure of these sequences is critical to anticipation, because stochastic arrival times (the defining characteristic of a point process) make sampling from interleaved sequences intractable, unless they are interleaved in a careful way. In particular, the elements of the control sequence must be interleaved at stopping times (a concept borrowed from the theory of stochastic processes).
>
> > Third, the part about "stoping time" seems unnecessarily complicated. Some stochastic calculus terms (e.g., "filtration") are introduced but do not really help readers understand the essence.
>
> We empathize with this feedback. We searched exhaustively for an elementary definition of a stopping time that avoids the technical machinery of filtrations; if such a definition exists, we would welcome a reference, and will happily use it in place of the current definition. In recognition that this definition is complicated, we provide a motivating discussion of stopping times in the introduction to Section 3. The formal discussion of stopping times themselves (and filtrations) is entirely confined to Section 3.1. The first paragraph of Section 3.1 attempts to provide additional intuition for the definition of a stopping time. Because stopping times are central to our proposed method, we do believe that it is important to include the formal definition in the paper.

---

> > ### Comment · Reviewer_9MMW · 2024-02-11
> > **rebuttal misses my key points (continued)**
> >
> > re. model vs. data
> >
> > There are cases where you use "process" right, and there are cases where you use it wrong. I was simply pointing out when you use it wrong and request you to fix them. Similarly, LMs are distributions over text strings, then certainly we cannot use "LM" to describe a piece of text, can we?
> >
> > re. your key technique
> >
> > By "tokenize the seq", I encompass your related techniques to do the anticipation. Certainly, I understand the point of the paper is to do anticipation, like I write down in the Strengths part of my rebuttal. I think it is a clever idea, and I LIKE it.
> >
> > My this point is still about the connection you draw to point process.
> >
> > re. stoping time and related
> >
> > Please correct me if I am wrong. The stopping time (index) is just the time (index) of next observed event (control), is it right? Your sampling needs to be stopped because you have reached a control, which also means that you should start your sampling for the next interval. Is it right? If yes, then why not just introduce it that way? I feel that the "filtration" part is not necessary.

---

> > > ### Author Response · Authors · 2024-02-14
> > > **clarifying a misunderstanding**
> > >
> > > > Please correct me if I am wrong. The stopping time (index) is just the time (index) of next observed event (control), is it right? Your sampling needs to be stopped because you have reached a control, which also means that you should start your sampling for the next interval. Is it right?
> > >
> > > There is a misunderstanding. We will attempt to clarify below.
> > >
> > > > The stopping time (index) is just the time (index) of next observed event (control), is it right?
> > >
> > > There is an important nuance here, which is that the index of a control in a sequence is random: it depends upon upon the previous randomly-generated events in the sequence. So, e.g., if the control should be anticipated at time s=11, and we generate 4 events before generating a 5th event that exceeds time s=11, then the control will be placed at index 6. Alternatively, if we generate 7 events before generating an 8th event that exceeds time s=11, then the control will be placed at index 9.
> > >
> > > A "stopping time" can be thought of as a rule for the placement of controls that can be determined based on the history of events (paraphrasing the informal definition that we give at the beginning of Section 3.1). So in the case above, the rule is that we place a control anticipated at time s=11 immediately _after_ observing an event that exceeds time s=11. In contrast, suppose our rule was that we place a control in sort order: in that case, the control appears immediately _before_ the first event that exceeds time s=11. We can create sorted training sequences, but we cannot run inference on a model trained this way because
> > >
> > >   * we won't know until we generate the next event whether it exceeds time s=11, but
> > >
> > >   * we can't generate the next event until we know whether the control should come first.
> > >
> > > The rule (1) "a control on time s=11 appears immediately after the first event exceeding time s=11" (anticipatory ordering) is a stopping time (inference is tractable!). The rule (2) "a control on time s=11 appears immediately before the first event exceeding time s=11" is not a stopping time (inference is intractable!). We wrote this paper in part because rule (1) (the sort order) is such a natural thing to try, but surprisingly inference for a model of these sequences doesn't work. What is needed is a rule like (2) (anticipatory order) and the formal name for rules like this one (for which inference works) is a "stopping time."
> > >
> > > Please let us know if this helps! We are happy to provide further clarification.
> > >
> > > We hope that the discussion above also helps to clarify why we need to introduce the language of temporal point processes in our paper (and we apologize if our earlier response to your comments about tokenization and the LM objective seemed curt). The anticipatory ordering of sequences depends upon the temporal ordering of events and controls in a subtle way: we aren't simply serializing the events and controls according to time-stamps (that would be the sort order, which doesn't work). We need to define the temporal structure of events and controls in order to carefully describe how time stamps in these two processes map to indices in the sequences that we ultimately train a model on. We also need to book-keep the time stamps of events and controls (not just their sequence indices) during inference in order to applying the rule (stopping time) for when to insert control tokens. So it really is important to discuss that events and controls are sampled from temporal point processes.

---

> > > > ### Comment · Reviewer_9MMW · 2024-02-19
> > > > **"stoping time" clarified**
> > > >
> > > > Thanks for your detailed clarification.
> > > >
> > > > Yeah, I understand and I think we are on the same page.
> > > >
> > > > So the key presentation issue here is to clarify the nuance, and introducing fancy concepts like "filtration" doesn't really do any help, right?
> > > > I think the space for "filtration"-related things should be just contributed to clarifying what "stoping time" precisely means.

---

> > > > > ### Author Response · Authors · 2024-02-20
> > > > > **how can we help?**
> > > > >
> > > > > > I think the space for "filtration"-related things should be just contributed to clarifying what "stoping time" precisely means.
> > > > >
> > > > > We have tried to do this, but we're open to suggestions about how to do it better. The formal discussion of stopping times is limited to Section 3.1, and the first paragraph of this section provides an informal explanation of the concept, with examples. Filtrations are mentioned just twice: once in the definition of a stopping time, and once in the following paragraph that describes how the definition is applied in our context.
> > > > >
> > > > > Would additional sign-posting help? Section 3.1 is self-contained: we could add a comment clarifying that Section 3.1 draws a connection to a classic mathematical concept, but isn't needed to understand the definition of the method itself.

---

> ### Author Response · Authors · 2024-02-19
> **re: modeling long sequences**
>
> > How did your method handle long sequences?
>
> A model architecture has a limited budget of "context" for processing tokens: this is the case for both the Transformers trained in our paper, and for LSTM (Mei's architecture) which--when sequence lengths are long--are trained fixed-length subsequences using truncated back-propagation through time. Given a limited context, anticipation ensures that the model's context is allocated to:
>
>     (1) all the (anticipated) controls that will arrive in the next delta seconds, and
>     (2) the most recent controls and events that have arrived in the past.
>
> > Did you do anything specific (sorry if I miss it) to make your method scalable to long sequences but it is not done by Mei et al. 2019?
>
> Mei et al. 2019 do not consider the setting where sequence length > model context, and offer no suggestions about how to approach modeling in this setting. Anticipation is what allows our model to make predictions when there are far more tokens in the sequence than the model can process, by ensuring that the model's context always contains the recent event history (unidirectional context) and both recent and near-future controls (limited bidirectional context).
>
> We allude to these challenges in our discussion of long sequences and locality (e.g., paragraph 3 of Section 1, and paragraph 1 of Section 3). We will elaborate on this discussion, drawing from the comments above, with an explicit description of how anticipation allocates model context when sequence length > model context.

---

> > ### Comment · Reviewer_9MMW · 2024-02-20
> >
> > I agree that your rearrangement of tokens may take a better use of the context, but it relies on the locality, right?
> >
> > But that difference doesn't seem fundamental because other approaches (e.g., SMC) are not necessarily bad after truncation because they also have some locality.
> >
> > E.g., you can truncate 12K tokens into pieces of 400 tokens and run a SMC method, which may work bad at the ends of each segment but find in the middle; if your segments have some overlap, then the end issue may also be solved.
> >
> > I guess I am not suggesting doing an awful amount of work to just compare with this baseline.
> > I just mean that you can apply the same treatment that you have done for your approach, and it doesn't seem a real issue to me (though I may be wrong).

---

### Author Response · Authors · 2023-12-14
**Author Response to Reviews**

**Edited 01/19/2024.** We have updated this general response to reflect the additional review.

We thank the reviewers for their feedback on our paper.

We remark that this paper makes two substantive contributions: a methodological contribution to the conditional generative modeling of temporal point processes (anticipation) and an empirical contribution to the development of generative models for music. All three reviews express appreciation for both of these contributions. Regarding our methodological contribution, they praise the clarity of our writing (Wdtb), the choice of examples used to describe this method (ynfy), and the overall cleverness of the method (9MMW). Regarding the music application, the reviews note that our method and model are superior to baselines (Wdtb), that our human evaluation is clearly described (ynfy), and that the results are “compelling” (9MMW).

The reviews raise questions about broader applications of anticipation beyond the music domain (Wdtb), our focus on human evaluation (Wdtb) and comparison to the original Music Transformer’s accompaniment experiment (ynfy). There is disagreement about the clarity of our presentation, with both praise for our exposition (Wdtb, ynfy) as well as criticism (9MMW). We address these questions in our individual responses to the reviewers.

---

> ### Comment · Action_Editor_bWE8 · 2023-12-20
>
> Note that a third reviewer dropped out late, and I am looking for another reviewer to take a look at the paper.  This may push things back a bit.

---

> > ### Author Response · Authors · 2023-12-20
> > **Message Received**
> >
> > Okay, thank you for letting us know.

---

### Author Response · Authors · 2024-03-05
**Updated manuscript**

In response to discussions with the reviewers, we have updated a revised version of our paper. Changes from the previous version are highlighted in red in the manuscript, and enumerated below.

In light of our discussion with Reviewer 9MMW:

  * Overall: additional care in our use of language regarding temporal point processes,
  * Section 3: elaboration of the example illustrating the difference between sort order and anticipatory order,
  * Section 4: added discussion of modeling in the regime where sequence length > model context length,
  * Section 5: added discussion of sequential monte carlo methods for infilling.

In light of our discussion with Reviewer Wdtb:

  * Section 6: added comments on the broader applicability of anticipation,
  * Appendix H: added quantitative metrics reported in this discussion.

The previous revision (and this one) reflect our earlier agreement with Reviewer ynfy to:

  * combine discussion of related work into a single section,
  * discuss the Music Transformer's seq2seq model.

We again thank the reviewers for their feedback and engagement with our work.

---

### Decision · Action_Editor_bWE8 · 2024-03-15

**Recommendation:** Accept with minor revision

**Comment:**

Thank you for providing manuscript updates.  I had the reviewers take a look at the latest manuscript update, particularly the one negative reviewer.  This reviewer still has some concerns, namely about some of the technical writing, and maintained their lean reject recommendation.  The other two reviewers recommended accepting the paper.  I am going to paste the comment the negative reviewer made below regarding rigor in the manuscript:

"I am confused why the authors still want---so much---to relate [their method] to temporal point process.

Their new text in Related Work says: "Sequential Monte Carlo (SMC) samplers have been considered for infilling of both continuous temporal point processes (Mei et al., 2019) and sequence models (Lin & Eisner, 2018); the later more directly relates to our setting because discretizing time (Section 2.1) reduces point process modeling to sequence modeling."

However, the former (Mei et al., 2019) is on point processes while the latter (Lin & Eisner, 2018) is on discrete-time sequences (like languages). Then it is very strange that the authors relate their work to the latter but still calls their model a "point process". For this reason, I hold my opinion that the technical arguments are lack of rigor."

The reviewer also elsewhere noted that "a lot of unnecessary math complication (e.g., filtration) should be removed."  It would be very helpful to address these concerns, but I don't see this as fundamentally problematic for publishing the paper; I think it would just clarify the contributions and clean up the presentation.  I would like to see a minor revision addressing these issues; this should not require too much additional work.

**Audience:**

Yes I think the TMLR audience would be interested.  One of the reviewers noted in their recommendation that they had concerns of the TMLR audience "being convinced of the broader significance of this work in the field."  However, overall this reviewer also felt fairly strongly that the paper was suitable and should be accepted.  Another reviewer felt that the paper could potentially be of more interest to the community if it were to be broadened beyond symbolic music; however, I think that the application to symbolic music is sufficient and interesting, as there is definitely a community within the broader TMLR community working specifically in this space.

**Claims And Evidence:**

The experiments presented on MIDI were considered sufficient by the reviewers.  During the initial reviews, all three reviewers asked for additional results, but all of these were resolved through the discussion period and updates to the manuscript.  One of the reviewers noted that additional experiments beyond the symbolic music setting would have helped, but this seems to me to be out of scope for the current paper.  Overall, it appears that the reviewers are satisfied with both the claims made by the paper (modulo some presentation issues noted in the comments below) as well as the empirical evidence provided.